# Mapping lung cancer epithelial-mesenchymal transition states and trajectories with single-cell resolution

Loukia G. Karacosta[1,2], Benedict Anchang[1,2], Nikolaos Ignatiadis[3], Samuel C. Kimmey [4], Jalen A. Benson[5], Joseph B. Shrager[5], Robert Tibshirani[1,3], Sean C. Bendall [4,6] & Sylvia K. Plevritis[1,2,6]*

Elucidating the spectrum of epithelial-mesenchymal transition (EMT) and mesenchymal-epithelial transition (MET) states in clinical samples promises insights on cancer progression and drug resistance. Using mass cytometry time-course analysis, we resolve lung cancer EMT states through TGFβ-treatment and identify, through TGFβ-withdrawal, a distinct MET state. We demonstrate significant differences between EMT and MET trajectories using a computational tool (TRACER) for reconstructing trajectories between cell states. In addition, we construct a lung cancer reference map of EMT and MET states referred to as the EMT-MET PHENOtypic STAte MaP (PHENOSTAMP). Using a neural net algorithm, we project clinical samples onto the EMT-MET PHENOSTAMP to characterize their phenotypic profile with single-cell resolution in terms of our in vitro EMT-MET analysis. In summary, we provide a framework to phenotypically characterize clinical samples in the context of in vitro EMT-MET findings which could help assess clinical relevance of EMT in cancer in future studies.

[1] Department of Biomedical Data Science, Stanford University, Stanford, USA. [2] Department of Radiology, Stanford University, Stanford, USA. [3] Department of Statistics, Stanford University, Stanford, USA. [4] Department of Pathology, Stanford University, Stanford, USA. [5] Department of Cardiothoracic Surgery, Stanford University, Stanford, USA. [6] These authors jointly supervised this work: Sean C. Bendall, Sylvia K. Plevritis. *email: sylvia.plevritis@stanford.edu

Malignant cells often hijack biological processes that are utilized by their normal cell counterparts[1]. One example is the epithelial–mesenchymal transition (EMT), a developmental program critical to embryogenesis and wound healing[2,3]. During EMT, epithelial cells undergo dramatic biochemical and morphological changes and acquire migratory and stem-like traits[4]. EMT is a dynamic process and under specific conditions is reversible (mesenchymal–epithelial transition, MET), highlighting a phenotypic plasticity that has been observed in both normal and malignant cells[3].

The clinical significance of EMT in cancer is well documented, although controversial[5]. Mesenchymal signatures are indicators of poor prognosis in various adenocarcinomas, including lung[6,7]. In addition, several studies have demonstrated the role of EMT in drug response and resistance[8,9], offering evidence that EMT signatures can be of therapeutic value. However, because most of these cancer-related EMT studies are based on bulk gene expression data from clinical specimens, it is often unclear whether clinical EMT signatures originate from mesenchymal malignant cells as opposed to tumor stromal cells (e.g., fibroblasts), which express EMT canonical markers. Furthermore, malignant cells with a purely mesenchymal phenotype are considered rare, with a small chance of clinical observation[10]; thus, the existence of such malignant cells is debated.

Adding to the complexity in understanding the clinical significance of EMT is the recognition that EMT is not a binary process (strictly defined by epithelial and mesenchymal states), but instead a spectrum of states where transitioning cells exhibit partial EMT phenotypes with both epithelial and mesenchymal features. Partial EMT phenotypes have been observed in clinical cancer specimens and negatively correlate with survival[9,11], but are poorly understood. Recent studies have attempted to better define EMT states using single-cell approaches; however, they were primarily focused on preclinical models or clinical samples without bridging the two. Pastushenko et al.[12] demonstrated the existence of partial EMT states in mammary and skin cancer by examining a large number of surface markers with flow cytometry and single-cell RNA-sequencing (sc-RNAseq). Gonzalez et al.[13] identified partial EMT states in ovarian cancer specimens with mass cytometry. While these studies provide important insights, they did not directly relate their findings to EMT states that have been well characterized in preclinical in vitro reference models. On the other hand, mass cytometry was used to study drug perturbations on tumor growth factor-β (TGFβ)-induced EMT in mouse epithelial cancer cells[14], and sc-RNAseq was used to study TGFβ-induced EMT in human breast epithelial cells[15]; however, neither of these studies provided a means to assess the clinical relevance of their respective findings.

In this study, we use high-dimensional single-cell analysis to identify and characterize EMT states observed in lung cancer clinical specimens in terms of the spectrum of states observed in well-established lung cancer cell lines. Specifically, through mass cytometry time-course analysis of TGFβ-modulated EMT and MET in lung cancer cells, we computationally define eight EMT and MET states with which we create an EMT–MET PHENOtypic STAte MaP (PHENOSTAMP). In addition, we develop and apply the TRAjectory of CElls Reconstruction (TRACER) algorithm, which compares state transitions in EMT and MET. Finally, we develop and apply a machine learning algorithm to project malignant cells from clinical samples onto the EMT–MET PHENOSTAMP in order to assess clinically relevant EMT and MET states in terms of our in vitro time-course analysis. Use of the EMT–MET PHENOSTAMP, informed by our trajectory analysis, provides a means to distinguish whether cells are undergoing EMT or MET and thereby has the potential to identify EMT- or MET-related phenotypic cell properties such as state-specific differential drug sensitivities[9,16].

## Results

**Identifying canonical EMT states in lung cancer cells.** To identify EMT states in lung cancer cells, we used TGFβ for EMT induction. First, we examined TGFβ responsiveness in three non-small-cell lung carcinoma (NSCLC) cell lines (HCC827, A549, H3255) towards bulk expression of widely accepted EMT markers (E-Cadherin, Vimentin, CD44[2,4,17]) (Fig. 1a). A549 cells displayed partial EMT (pEMT) characteristics before TGFβ treatment (basal co-expression of epithelial (E-Cadherin) and mesenchymal (Vimentin, CD44) markers). With treatment, A549 cells exhibited loss of E-Cadherin and acquired mesenchymal-like morphology (Fig. 1a, Supplementary Fig. 1). H3255 cells exhibited strong basal epithelial features and retained their epithelial morphology and protein expression with TGFβ treatment without displaying EMT-related changes (Fig. 1a, Supplementary Fig. 1). The HCC827 cell line displayed epithelial features at basal conditions and underwent dramatic EMT marker and morphology changes with treatment (Fig. 1a, b), and hence was chosen as the exemplary cell line for our EMT studies.

Morphological heterogeneity was observed in HCC827 cells during EMT, as evidenced by confocal imaging (Fig. 1b). Intriguingly, cells that were not incorporated in epithelial islands expressed Vimentin even in the absence of exogenous TGFβ, demonstrating that cell density likely affects EMT status (Fig. 1b (b)). This finding was corroborated when we examined EMT in HCC827 cells with flow cytometry under varying seeding conditions that controlled confluency (Supplementary Figs. 1 and 2) and is consistent with previous reports for normal epithelial cells[18]. Transition was also reflected by CD44 and CD24 gradual changes (stemness markers often used in breast cancer EMT studies), patterning four readily observed gating populations (Supplementary Fig. 1). These subpopulations—referred to as "states"—were characterized by progressive increase of CD44 and decrease of CD24 in only a small subset of cells (CD44$^{hi}$/CD24$^{lo}$), which represents a phenotypic state regarded as cancer stem cell-like (CSC-like)[4,19].

To adequately capture EMT states, we optimized a TGFβ time-course during which we kept confluency consistent by re-seeding cells at each experimental time-point. Cells efficiently underwent EMT within 10 days of TGFβ treatment towards morphology (Fig.1c, top images) and changes in E-Cadherin, Vimentin, CD44 and CD24 expression (Fig. 1c, biaxial plots). Since these markers are commonly used for EMT characterization, we refer to these four states, as canonical EMT states. As CD44/CD24 biaxial plots gave four well-separated populations for manual gating (Supplementary Fig. 1), we used these to assess E-Cadherin/Vimentin expression (Fig. 1c, bottom plots) and designated the four canonical states into (i) an epithelial (E) (E-Cadherin$^+$/Vimentin$^-$), (ii) a pEMT (E-Cadherin$^+$/Vimentin$^+$), (iii) a mixed pEMT/M (E-Cadherin$^+$/Vimentin$^+$, E-Cadherin$^-$/Vimentin$^+$), and (iv) a small mesenchymal CSC-like state (E-Cadherin$^-$/Vimentin$^+$/CD44$^{hi}$/CD24$^{lo}$), denoted here as M* (Fig. 1d, Supplementary Fig. 1). Each state's occupancy with time is depicted in Fig. 1e. Mean fold expression of the four markers recapitulated the observed changes towards both control conditions with time and towards the E state when examined in each state separately (Fig. 1f). Unexpectedly, E-Cadherin transiently increased in the pEMT state. We speculate this may be due to increased surface interactions between cells of enlarged size observed during initial EMT stages (Supplementary Fig. 1). We also observed time-dependent changes in the EMT transcription factor Twist[20] (Fig.1f). Specifically, Twist expression

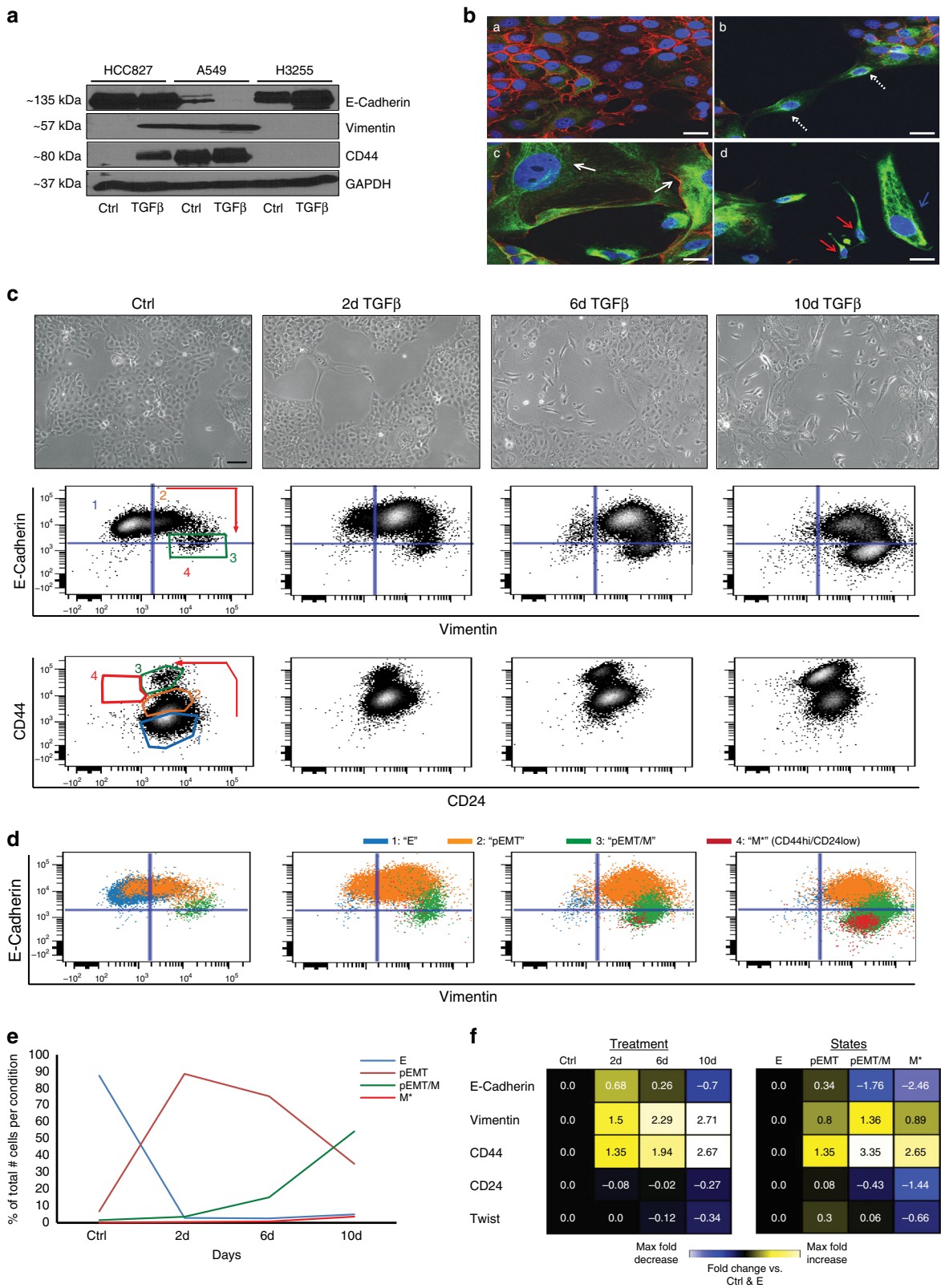

was highest in the pEMT state and lowest in the M* state (Fig. 1f, right).

**Phenotyping canonical EMT states with mass cytometry.** Manual gating based on the expression of just four markers was helpful for our initial assessment of canonical EMT states, albeit this approach does not fully characterize the heterogeneity of these states. For a more thorough characterization of the canonical EMT states, we performed a mass cytometry time-course experiment on HCC827 cells where TGFβ was added for 10 days and subsequently withdrawn to observe EMT and MET,

**Fig. 1 Identifying canonical EMT states in lung cancer cells through TGFβ time-course analysis. a** Immunoblot of EMT markers in the presence or absence of TGFβ in three NSCLC cell lines (2 week treatment in HCC827 and H3255 cells, 1 week treatment in A549 cells). **b** Representative confocal images of HCC827 cells stained for E-Cadherin (red) and Vimentin (green) in the absence (a, b) or presence of TGFβ (c, d) for 10 days. Magnification ×40, scale bar 20 µm. White dotted arrows indicate cells that show loss of E-Cadherin and gain of Vimentin expression in untreated conditions. White solid arrows show enlarged cells on the periphery of epithelial islands that have acquired a pEMT phenotype (E-Cadherin+/Vimentin+); blue and red solid arrows indicate either enlarged and elongated or small and spindle-shaped cells that have acquired mesenchymal phenotypes (E-Cadherin−/Vimentin+). **c** Representative images of HCC827 cells treated with TGFβ (5 ng/mL) for 2, 6, and 10 days under continuous re-seeding conditions (see also Supplementary Figs. 1 and 2 and Methods). Magnification ×10, scale bar 200 µm. Below each image are shown respective E-Cadherin/Vimentin and CD44/CD24 flow cytometry plots. Arrows indicate changes in marker expression during EMT and numbers 1–4 designate the identified canonical EMT states, respectively: Epithelial (E), pEMT, pEMT/Mesenchymal (M), and M* (characterized by CD44hi/CD24lo, CSC-like cells). **d** Color-coded gated EMT canonical states depicted on E-Cadherin/Vimentin flow cytometry plots per experimental condition (see also Supplementary Fig. 1). **e** EMT state dynamics during TGFβ time course. Color-coded EMT states were calculated and depicted as % of total number of cells at each experimental time-point. **f** Heatmap summary of EMT marker fold (arsinh) changes relative to control condition when analyzed in all cells (left) and relative to the E state when analyzed in the four canonical EMT states individually at the 10-day TGFβ time-point (right). Source Data are provided as a Source Data file.

respectively (Fig. 2a). For optimal confluency conditions, at each time-point, cells were collected for both mass cytometry analysis and re-seeding for subsequent time-points. In total, 28 markers were chosen to characterize EMT states as well as proliferative, signaling, and apoptotic cell status (Fig. 2a, Supplementary Table 1). The four canonical EMT states were reproduced on mass cytometry (Fig. 2b, Supplementary Figs. 2 and 3) as were the fold expression trends of all previously discussed markers (Fig. 2c, Supplementary Fig. 3). Upon TGFβ withdrawal, a proportion of cells that had undergone EMT returned to an epithelial state, an observation supported by canonical EMT marker expression and morphological assessment (Fig. 2b, c).

Mass cytometry enabled us to examine dynamic changes of additional markers (Fig. 2d, Supplementary Fig. 3). Keratins and other epithelial cell surface markers (e.g., MUC1, TROP2) showed expected decreases during EMT and subsequent return to basal levels following MET. Most signaling and proliferative markers rapidly increased in expression on day 2 under TGFβ treatment, and subsequently decreased by day 10 in TGFβ. Decreased expression of these markers was sustained during the first 2 days of withdrawal, time-points for which M and M* cells were at highest numbers. Specifically, this pattern was observed for pSMAD2/3, a key TGFβ signaling molecule[21], as well as for pEGFR (epidermal growth factor receptor) and pS6, suggesting that signaling is downregulated in the M state. Notably, pEGFR and pS6 were lowest in the CSC-like M* cells (Supplementary Fig. 3), corroborating studies that have associated EMT with tyrosine kinase inhibitor (TKI) resistance[8] and low p-mTOR (mammalian target of rapamycin) activity with stem cell maintenance[22]. We also observed an increase of PD-L1 during EMT, substantiating reports that link EMT with tumor immune evasion[23]. Among the measured EMT transcription factors, apart from the transient activation of Twist, we did not detect significant Snail or Slug expression. However, Oct3/4 and Nanog showed a substantial increase throughout the EMT time-course, consistent with a lung adenocarcinoma study where co-expression of Oct3/4 and Nanog were found to be critical for EMT[24].

**Defining computationally derived EMT and MET states**. Our above-described assessment of EMT states showed that at least at the protein level cells seem to transition through a spectrum of overlapping phenotypes. Given the high dimensionality of our data, we applied unsupervised cluster analysis to computationally derive well-defined EMT and MET states. From hereon, the previously described canonical EMT states hold no particular significance, but will be related to computationally derived EMT states. We first applied CCAST, a clustering algorithm that

applies non-parametric tests to partition the data in the form of a decision tree on the pooled mass cytometry data across all time-points[25]. Clustering markers E-Cadherin, Vimentin, CD44, CD24, MUC1, and Twist were selected based on principal component analysis (PCA) of the data (PCA and principal component loadings shown in Supplementary Table 2 and Supplementary Fig. 4, respectively). CCAST identified eight most prominent states, each state having ≥1% of total number of cells analyzed. The CCAST decision tree and downstream analyses, as well as the relation between the computationally derived with the four canonical EMT states are shown in Supplementary Figs. 5 and 6.

Heatmaps of the six clustering markers within each of the eight states revealed minimal within-state marker heterogeneity compared to between-state marker heterogeneity (Fig. 3a). We labeled the derived EMT and MET states based on the expression profile (e.g., E-Cadherin+/Vimentin− in epithelial states, E-Cadherin+/Vimentin+ in pEMT states, etc.) and respective time-course pattern, which helped distinguish, for example, mesenchymal cells from cells undergoing MET (i.e., cells appearing after TGFβ removal). Specifically, we identified three E states (E1, E2, E3), three pEMT states (pEMT1, pEMT2, pEMT3), one M state, and one MET state (Fig. 3b). States E1, E2, and E3 displayed low expression of Vimentin and CD44, but varying degrees of E-Cadherin, CD24, and MUC1 expression. All E states were highest in numbers at time 0 and dramatically decreased with TGFβ treatment; only E1 and E2 rebounded following TGFβ withdrawal. States pEMT1, pEMT2, and pEMT3 co-expressed E-Cadherin and Vimentin, consistent with a pEMT phenotype. pEMT2 and pEMT3 specifically included a subgroup of Twist+ cells (Fig. 3a, boxed regions). All pEMT states emerged upon TGFβ treatment, decreased with continued TGFβ treatment, and with the exception of pEMT3, transiently re-emerged during withdrawal. State M was characterized by loss of epithelial markers (E-Cadherin, MUC1), and high Vimentin and CD44 expression, and was highest in numbers at day 10 in TGFβ. Interestingly, Twist+ cells were not detected within this state, confirming a study that found that CSC-like properties arise following transient Twist activation during EMT[26]. Although our analysis did not distinguish CSC-like cells as a separate cluster, interrogation of other markers in cells expressing the lowest CD24 levels (e.g., pS6) helped us define the CSC-like cells within the M state (M*, Supplementary Fig. 6). State MET, perhaps most interesting in our analysis, peaked upon withdrawal specifically, representing cells that had perhaps initiated MET. The MET state differed from the M state in only one of six clustering markers, exhibiting increased MUC1 expression. Of note, MUC1 has been reported to be a MET marker in nephrogenesis[27], corroborating our findings here in lung cancer. MET cells also expressed higher

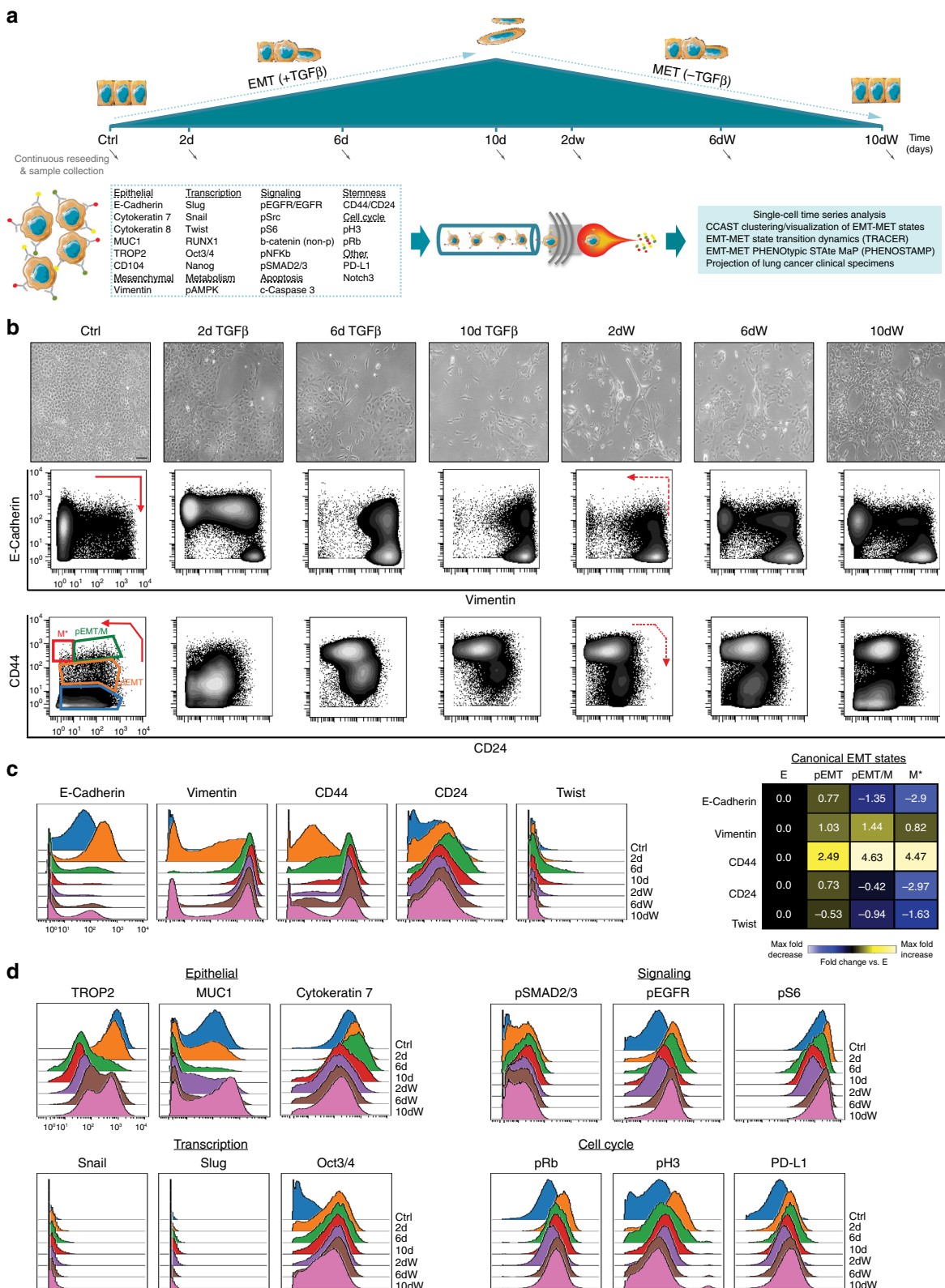

**Fig. 2 Mass cytometry time-series analysis offers deep single-cell resolution of canonical EMT states. a** Workflow schematic of cell culture conditions for TGFβ time-course (addition and withdrawal (W) time-points) and mass cytometry analysis. Arrows indicate times at which cells were collected for both analysis and re-seeding for the remaining time-points (see also Supplementary Table 1, Figs. 2 and 3, and Methods). **b** Representative images of HCC827 cells at each time-point (magnification ×10, scale bar 200 μm) with respective E-Cadherin/Vimentin and CD44/CD24 mass cytometry plots shown below. Arrows indicate the previously observed canonical EMT marker changes and states, respectively. **c** Histogram overlays illustrating EMT marker expression distributions in HCC827 cells undergoing EMT and MET. On the right, a heatmap summary of the canonical EMT marker fold (arsinh) changes relative to state E when analyzed in the four EMT states individually at the 10-day TGFβ time-point. **d** Additional marker distributions in HCC827 cells undergoing EMT and MET (see Supplementary Fig. 3 for remaining markers and an independent biological replicate experiment).

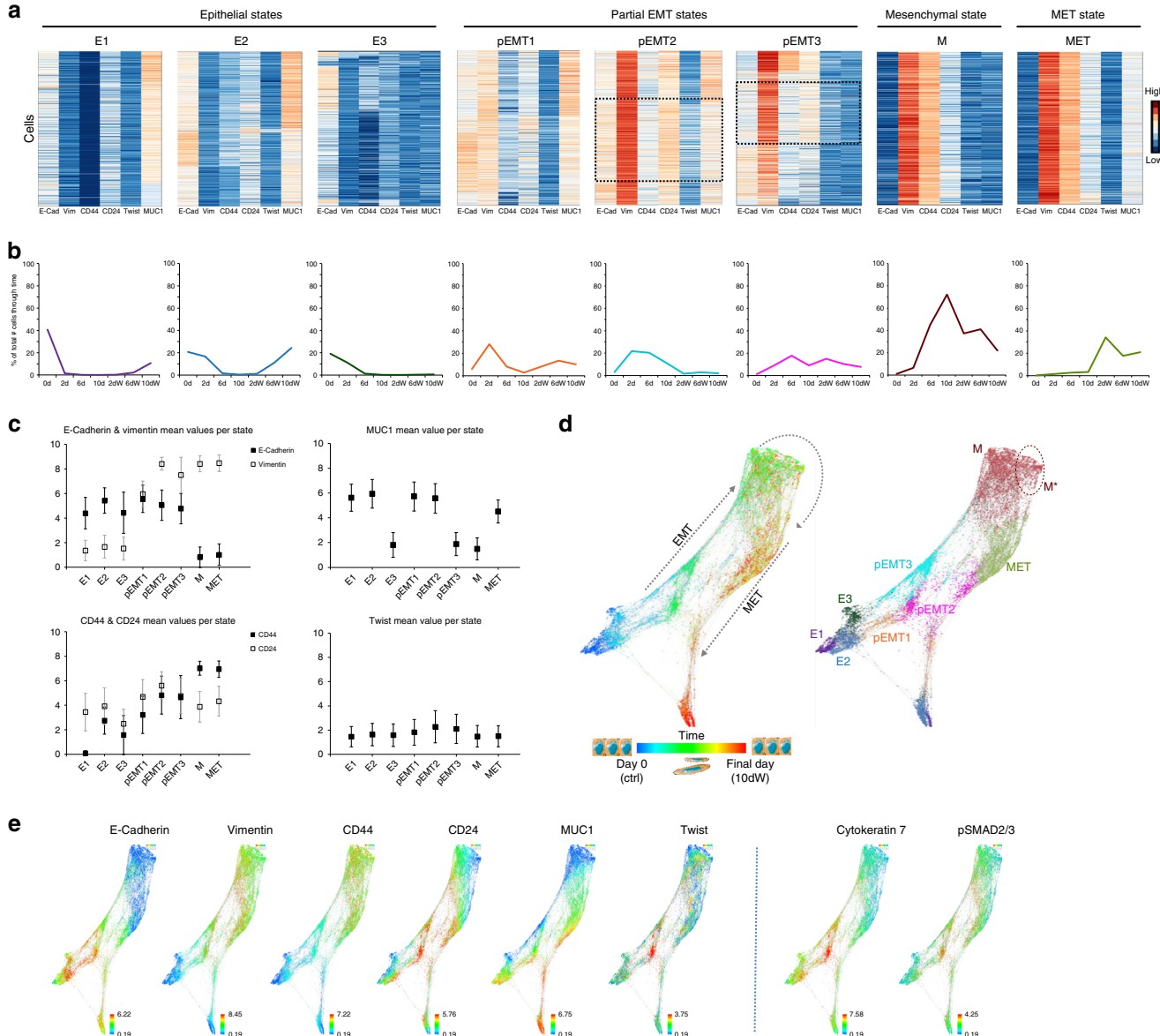

**Fig. 3 Unsupervised analysis of mass cytometry data reveals computationally derived EMT and MET states. a** Heatmaps depicting the expression of the six clustering markers in each cell (E-Cadherin, Vimentin, CD44, CD24, Twist, and MUC1) per computationally derived EMT and MET state (normalized global expression). Shown are the most prominent in number states that resulted from the CCAST algorithm performed on pooled mass cytometry HCC827 data. Dotted boxed regions illustrate subpopulations of Twist+ pEMT cells (see also Supplementary Fig. 6). **b** EMT and MET state dynamics. Each graph corresponds to the state heatmap above. States were calculated and depicted as % of total number of cells at each experimental time-point. **c** Graphs illustrating mean expression of the six clustering markers in EMT and MET states. Bars represent standard deviation (s.d.) of each marker within each downsampled state (E1 cells $n = 2394$, E2 cells $n = 3284$, E3 cells $n = 1316$, pEMT1 cells $n = 3359$, pEMT2 cells $n = 2943$, pEMT3 cells $n = 2663$, M cells $n = 9609$, MET cells $n = 3498$). **d** Time-resolved force-directed (FDL) layout of mass cytometry time-course data (left) and respective EMT and MET state annotations (right). Dotted area depicts subpopulation of M cells that exhibit CSC-like phenotypic characteristics (CD44hi/CD24lo, M*). **e** Time-resolved FDL layouts colored by protein expression of indicated markers (arsinh transformed data). See Supplementary Fig. 6 for remaining markers and analysis of an independent biological replicate experiment. Source Data are provided as a Source Data file.

levels of the epithelial markers TROP2 and cytokeratin 8 compared to M cells, confirming that these cells have initiated MET (Supplementary Table 3). Ordering the states according to temporal pattern and phenotype provided a quantitative examination of the expression changes of the six clustering markers (Fig. 3c). Similar results were observed in an independent biological replicate experiment (Supplementary Fig. 6).

To better visualize the phenotypic profiles and dynamic changes of the derived EMT and MET states, we constructed a force-directed layout (FDL) of the high-dimensional data using

Vortex[28] (Fig. 3d, Supplementary Fig. 6). Cells from each of the eight states were added sequentially per time-point and data were visualized in terms of time (Fig. 3d, left), or state (Fig. 3d, right). Thus, we were able to visually decipher the EMT from the MET trajectory, and identify states that appeared to be transient (e.g., pEMT3), or revisited during (e.g., pEMT1, pEMT2), or at MET completion (e.g., E1, E2). Strikingly, MET state appeared to be specific to the withdrawal time-points, supporting studies proposing that the MET trajectory differs from the EMT one[29,30]. Expression profiles of all markers are depicted in FDLs in Fig. 3e

and Supplementary Fig. 6. FDLs revealed that Twist[+] cells are characterized by high expression of cytokeratins 7 and 8, signaling molecules (e.g., pSMAD2/3, pEGFR), and transcription factors Oct3/4 and Nanog. High cytokeratin expression corroborates studies showing that Twist[+] cells constitute a specific type of epithelial cells[31]. On the other hand, cells in the M state are characterized by low signaling and proliferative profiles, a finding that agrees with our earlier observations (Fig. 2, Supplementary Fig. 3) and published reports[22,32].

**Constructing an EMT–MET PHENOtypic STAte MaP.** Apart from visualizing the phenotypic, time-dependent properties of EMT and MET states, our goal was to construct an EMT–MET PHENOtypic STAte MaP (PHENOSTAMP) for inferring the presence of the eight EMT and MET states in NSCLC clinical specimens with single-cell resolution. First, we generated a two-dimensional (2D) t-distributed stochastic neighbor embedding (t-SNE) projection of the eight states. Given that the EMT and MET states showed considerable phenotypic overlap on the t-SNE space, we employed Voronoi and Convex Hull analysis[33] to achieve density-driven segmentation of the t-SNE landscape; each segment corresponding to an identified EMT/MET phenotypic state at its peak during our time-course analysis (Fig. 4a, b and Supplementary Figs. 5 and 7). Analysis of cell density over time on the EMT–MET PHENOSTAMP confirmed our previous assessment that the MET trajectory differs from the EMT one and is specifically characterized by the appearance of the MET state at the 2-day withdrawal time-point (Fig. 4c). However, it also revealed the possibility that cells that belong to pEMT states exhibit plasticity that enables them to follow the same state trajectory back to E states under withdrawal. This observation is heuristically demonstrated in a density-driven preliminary assessment of transitions within the 2D map, depicted in a conceptual schematic model (Fig. 4d). When we applied the computational tool Slingshot[34] to these data, it independently resolved the trajectory that involves the appearance of the MET state during withdrawal (Supplementary Fig. 7). However, it did not fully capture our empirical observations of likely state transitions, given that as most pseudotime algorithms, it forces cells onto a deterministic trajectory derived from prior knowledge.

**TRACER analysis identifies distinct EMT and MET trajectories.** Given that EMT is likely characterized by plasticity among states and not a single trajectory, we developed a trajectory algorithm, TRACER (Supplementary Fig. 5 and Methods), that does not rely on pseudotime assumptions. We assumed that EMT and MET are classical Markov processes[35] with transition probabilities that are constant between the states within EMT and MET, but can differ when comparing the two. We applied our model separately for the EMT and MET time-points and, through bootstrap analysis, we generated a distribution of transition probabilities between states under EMT and MET, as shown in Fig. 4e. The resulting medoid networks for EMT and MET are provided in Fig. 4f. The bootstrap analysis shows less non-zero state transition probabilities under MET compared to EMT, demonstrating that the transition between M and MET states is unique to MET. Our analysis also shows that cells in the identified states may transverse a set of possible EMT trajectories representing perhaps differing state kinetics (differentially weighted EMT transition probabilities, Fig. 4f). These results demonstrate statistically significant differences between EMT and MET trajectories as well as bi-directionality (plasticity) between certain states (e.g., the transition probability from pEMT1 to pEMT2 during EMT is 0.33, whereas during MET the transition from pEMT2 to pEMT1 is 0.27).

**Projection of NSCLC cell lines onto the EMT–MET PHENOSTAMP.** Given the wide range of phenotypic, and presumably functional, EMT–MET states across NSCLC cell lines, we tested whether our EMT–MET PHENOSTAMP could serve as a reference to reconcile known heterogeneity between cell lines and experiments. To project independent samples onto PHENOSTAMP, we trained a neural net algorithm[36,37] that predicts the bivariate t-SNE outputs of our EMT–MET PHENOSTAMP in terms of the selected six clustering markers (Supplementary Fig. 5 and Methods). We verified the performance of the projection using lung cancer cell lines with known features. First, we projected independently analyzed TGFβ HCC827 samples, and these matched previously observed mappings (Fig. 5a, left). Next, we projected two NSCLC cell lines analyzed with mass cytometry (A549, H3255), given that we had prior knowledge of their EMT status (Fig. 1a). In agreement with our previous findings, A549 cells mapped onto pEMT and M regions in control conditions and extended into the M region upon TGFβ treatment (Fig. 5a, middle, Supplementary Fig. 8), and H3255 cells mapped mostly on E, and to a lesser degree on pEMT1 regions, with minimal changes upon TGFβ treatment (Fig. 5a, right, Supplementary Fig. 8). These map projections were consistent with morphological assessment and respective E-Cadherin/Vimentin and CD44/CD24 expression profiles (Fig. 5b, c).

**Phenotyping NSCLC clinical samples with EMT–MET PHENOSTAMP.** We proceeded to determine the spectrum of EMT and MET states in NSCLC clinical specimens with single-cell resolution by projecting them onto the EMT–MET PHENOSTAMP. Five fresh NSCLC adenocarcinoma samples were obtained immediately after patient resection under Institutional Review Board (IRB) approval and underwent immediate dissociation for single-cell suspension. For mass cytometry analysis, we used the antibody panel developed for our time-course analysis, augmented with antibodies to sort out immune (CD45[+]), endothelial (CD31[+]), and stromal fibroblast (FAP[+]) populations[13]. Thus, we were able to discriminate immune, endothelial, and stromal cells from cells that were of varying levels of cytokeratins 7 and 8 that were negative for CD45, CD31, and FAP (Supplementary Figs. 2 and 9). The specimens harbored different mutations and differentiation status (Fig. 6, Supplementary Table 4). We validated both the mass cytometry run and projection, by analyzing in parallel independent HCC827 samples as a control (Supplementary Fig. 9). Upon projection, all clinical specimens showed the presence of a variety of EMT states that agreed with their respective E-Cadherin/Vimentin expression profiles (Fig. 6, Supplementary Fig. 9). Reassuringly, given the single-cell resolution of the mapping function, each clinical sample demonstrated a continuous sweep of EMT states, as opposed to occupying disjoint regions. Case nos. 1, 2, and 3 (all EGFR mutated) mapped primarily onto E regions with some extent into the pEMT1 regions, and expressed, as expected, higher pEGFR levels (Supplementary Fig. 10). This mapping is consistent with EGFR-mutated cell lines HCC827 and H3255 in basal conditions and confirmed by morphological assessment (Supplementary Fig. 9). Case No. 3, harboring an additional TP53 mutation, broadly extended into the pEMT regions, with a small fraction of cells in the M and MET regions. Case nos. 4 and 5, harboring TP53 and KRAS mutations, respectively, showed the most pEMT features, occupying pEMT2 and pEMT3 regions (and to a lesser degree M), reassuringly similar to A549 cells that are also KRAS mutated (Fig. 5). Interestingly, case nos. 4 and 5, in comparison to case nos. 1–3, were graded as poorly differentiated and showed higher levels of immune infiltration (Fig. 6, Supplementary Fig. 9), suggesting the association of these clinical

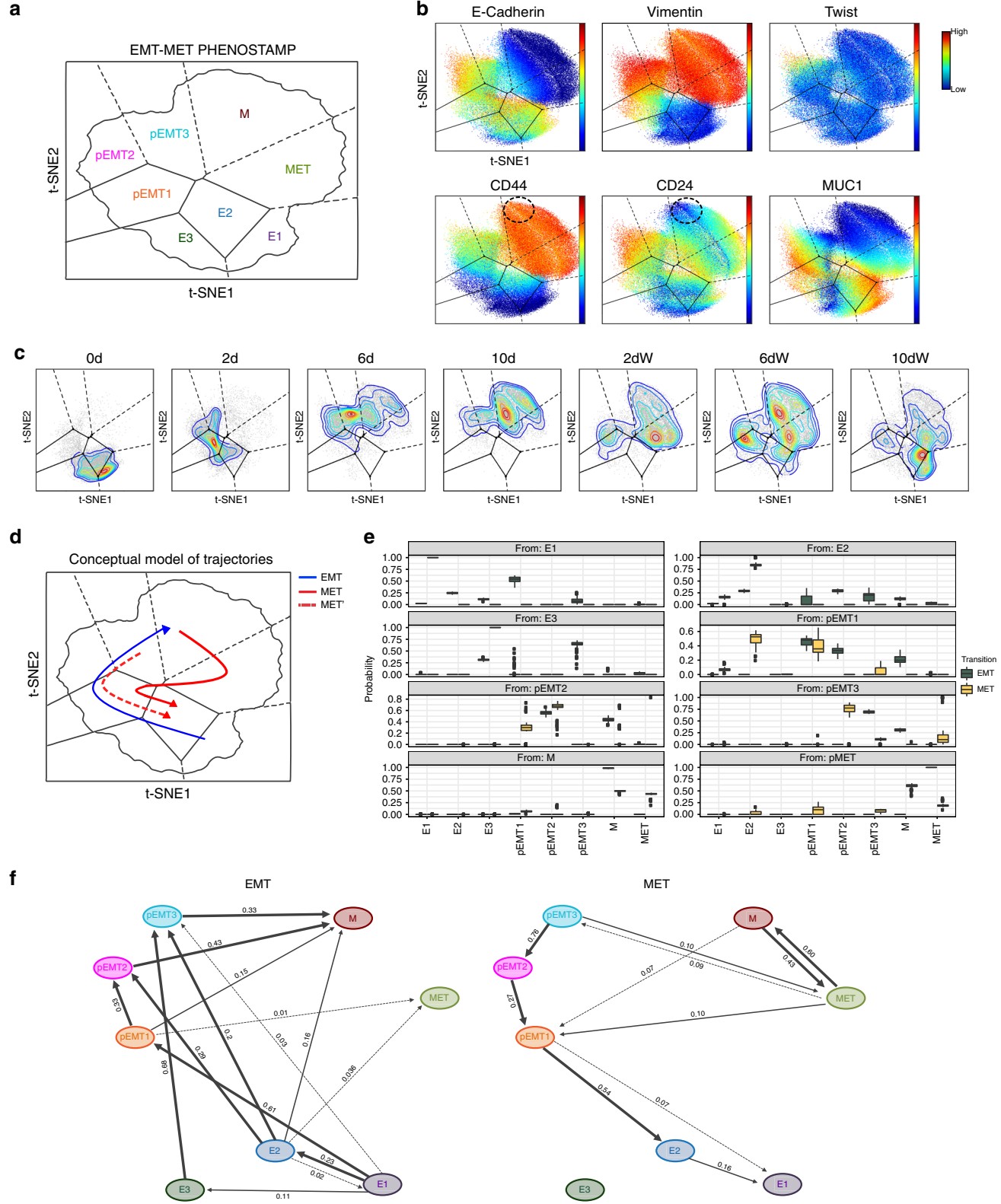

features with specific EMT states. While a larger sample size is necessary to solidify conclusions, these results are consistent with studies reporting correlations between NSCLC mutations and immune infiltration with EMT status[23,38,39]. Overall, we show that our EMT–MET PHENOSTAMP can be utilized to score and interpret clinical specimen data towards EMT and MET state heterogeneity and that similar approaches can be extended to

phenotyping a range of cellular processes that involve cell state transitions.

## Discussion

Elucidating what is believed to be a spectrum of EMT–MET states with single-cell resolution promises insights on their role in

**Fig. 4 Construction of the EMT–MET PHENOSTAMP and TRACER analysis identify distinct EMT and MET trajectories. a** Schematic of 2D t-SNE EMT–MET PHENOSTAMP. HCC827 CCAST state information was used to determine the highest-density areas of cluster-specific time-dependent bins and subsequent estimation of respective centers. Compartmentalization was achieved with Voronoi and Convex hull analysis. **b** Expression profiles of the 6 clustering markers in pooled HCC827 time-point data visualized on the EMT–MET PHENOSTAMP. Dotted circles represent CSC-like (CD44hiCD24lo) cells. See Supplementary Fig. 7 for remaining markers. **c** Time-point specific HCC827 t-SNE density plots. **d** Conceptual model of density plot-inferred EMT and MET trajectories of transitioning cells (see text for details). Blue arrow depicts the EMT trajectory, whereas red arrows depict two possible MET trajectories that both may take place depending on conditions; one utilizes the MET state, which supports the notion of hysteresis during MET (red solid line), and the other utilizes the previously visited pEMT states (red dotted line). **e** Bootstrap analysis comparing the distribution of time-independent state transitions as generated by the TRACER algorithm for EMT and MET, represented as box plots, each graph showing transition of a specific state to the all other states (x-axis). The centers of the box plots represent medians of the bootstrap transition distributions (bootstrap analysis based on $n = 10,000$ cells for each of seven measured time-points). The upper (lower) hinge shows the 75th (25th) percentile. The upper (lower) whisker extends from the upper (lower) hinge to the largest (smallest) value no further than 1.5 times the interquantile range. Box plots color coded as green (left) and yellow (right) represent EMT and MET, respectively. **f** Schematic diagram of the TRACER medoid network from bootstrap analysis, depicting transitional probabilities among states during EMT and MET. Arrows are weighted by probability strength. See Methods for further details. Source Data provided as a Source Data file.

---

cancer. We provide an integrated experimental–computational framework to define discrete lung cancer EMT and MET states and assess their clinical relevance. First, we identified EMT and MET states leveraging high-dimensional mass cytometry time-course analyses of lung cancer cell lines undergoing EMT and MET through TGFβ treatment and withdrawal. We then constructed EMT–MET PHENOSTAMP, and using a neural net algorithm, we projected clinical samples onto the map to evaluate their EMT and MET state profile with single-cell resolution. This integrated approach provides in vitro insights on EMT–MET biology and establishes a framework to translate in vitro observations to clinical samples.

The EMT process has been described as a spectrum of intermediate phenotypic states[40–42], and this is something we observe when we interrogate transitioning cells towards their protein expression changes. Cells appear to occupy a continuum of phenotypic space (biaxial plots in Figs. 1 and 2), making it difficult to discern discrete EMT populations. This is also evident by our clinical specimen projections (Fig. 6) as these occupy connected regions on our EMT–MET PHENOSTAMP. Other investigators have defined a discrete number of EMT states in various systems, albeit using either bulk gene expression data or surface-only protein markers[12,17]. In this study, we used CCAST analysis of mass cytometry time-course data to define eight distinct EMT–MET states at the single-cell proteomic level; thus, providing a well-defined EMT–MET state map that discretizes the EMT spectrum in order to gain an in-depth view of phenotypic transitions.

With our single-cell EMT–MET time-course analysis, we demonstrated not only the heterogeneity of EMT cell states but also transient properties that provide a deeper appreciation of the dynamism of EMT. For example, we showed that in NSCLC, epithelial states can be quite heterogeneous towards E-Cadherin, CD24, and, interestingly, MUC1 expression. Heterogeneity was also observed within the pEMT states for various markers, most notably Twist. Expression of EMT-specific transcription factors during EMT was recently reported to be dispensable for some cells, featuring an alternative EMT program that involved protein internalization[43]. We observed phenotypically distinct, transient Twist[+] cells that retained some epithelial identity and proliferative capacity, consistent with prior work[31]. Although transient Twist activity has been reported[26], these prior observations were made in cells where Twist was exogenously overexpressed; instead, we examined changes in *endogenous* Twist during EMT induction. Twist has been shown to be overexpressed in human lung adenocarcinoma and specifically correlated to EGFR mutations[44], as observed in two of three EGFR-mutated clinical samples we analyzed (Supplementary Fig. 9). Although we did

not detect other EMT-specific transcription factors (i.e., Slug, Snail, Zeb1, Fig. 2 and Supplementary Fig. 1), we cannot exclude the possibility that these may be activated in earlier EMT time-points not tested here.

Our time-course analysis of MET is a key aspect of our study. MET is thought to be critical for the establishment of secondary distant tumors. Yet, compared to EMT, MET is less studied, particularly with single-cell resolution. Some studies have shown that EMT is reversible among cells in pEMT states, but not necessarily among cells that have become mesenchymal, although this seems to be cell type dependent[45,46]. Even so, for cells undergoing MET, it is unclear whether the MET trajectory mirrors or differs from the EMT trajectory. Differing trajectories that we found is evidence of hysteresis, a phenomenon in which a future state depends on its history. Several mathematical modeling studies have provided evidence of hysteresis when comparing EMT and MET; however, these were based on gene expression or were not associated with specific phenotypic states[29,30]. By analyzing time-course data using TRACER, we found statistically significant evidence of hysteresis. In particular, we showed that some mesenchymal cells undergo MET utilizing a trajectory not observed under EMT and transit through a distinct identified state that we defined as MET. More specifically, our study supports two possible scenarios. In the first scenario, cells in the M state have undergone such significant (presumably epigenetic) changes that in order for some of them to undergo MET, they utilize a different trajectory. Of note, a significant proportion of cells failed to undergo MET after 10 days TGFβ withdrawal. It is possible that if we had prolonged withdrawal, more cells could have returned to E states, presumably through a combination of epigenetic/transcriptional mechanisms that regulate phenotypic switches[47]. Moreover, we found that if cells have not efficiently undergone EMT (majority of cells transition to pEMT rather than M states), most of them are able to undergo MET within 10 days of TGFβ withdrawal (Supplementary Fig. 7c). This observation is linked to the second scenario, in which, if during conditions that promote MET a cell is in a pEMT state, it utilizes a mirrored trajectory back to an epithelial state. Supporting this, TRACER detected bi-directionality between pEMT states (Fig. 4e, f). Notably, TRACER enables the interrogation of the bi-directional and plastic nature of EMT and MET processes, as opposed to pseudotime trajectory algorithms (e.g., Wanderlust, Monocle, Slingshot[34,48,49]) that are deterministic in nature, forcing the ordering of transitioning cells on a predefined developmental path. Specifically, TRACER utilizes the proportion of cells in each state per time-point to generate a distribution of transition probabilities, and presents more than one possible EMT trajectories (Fig. 4f). Nevertheless, TRACER's current limitation is that

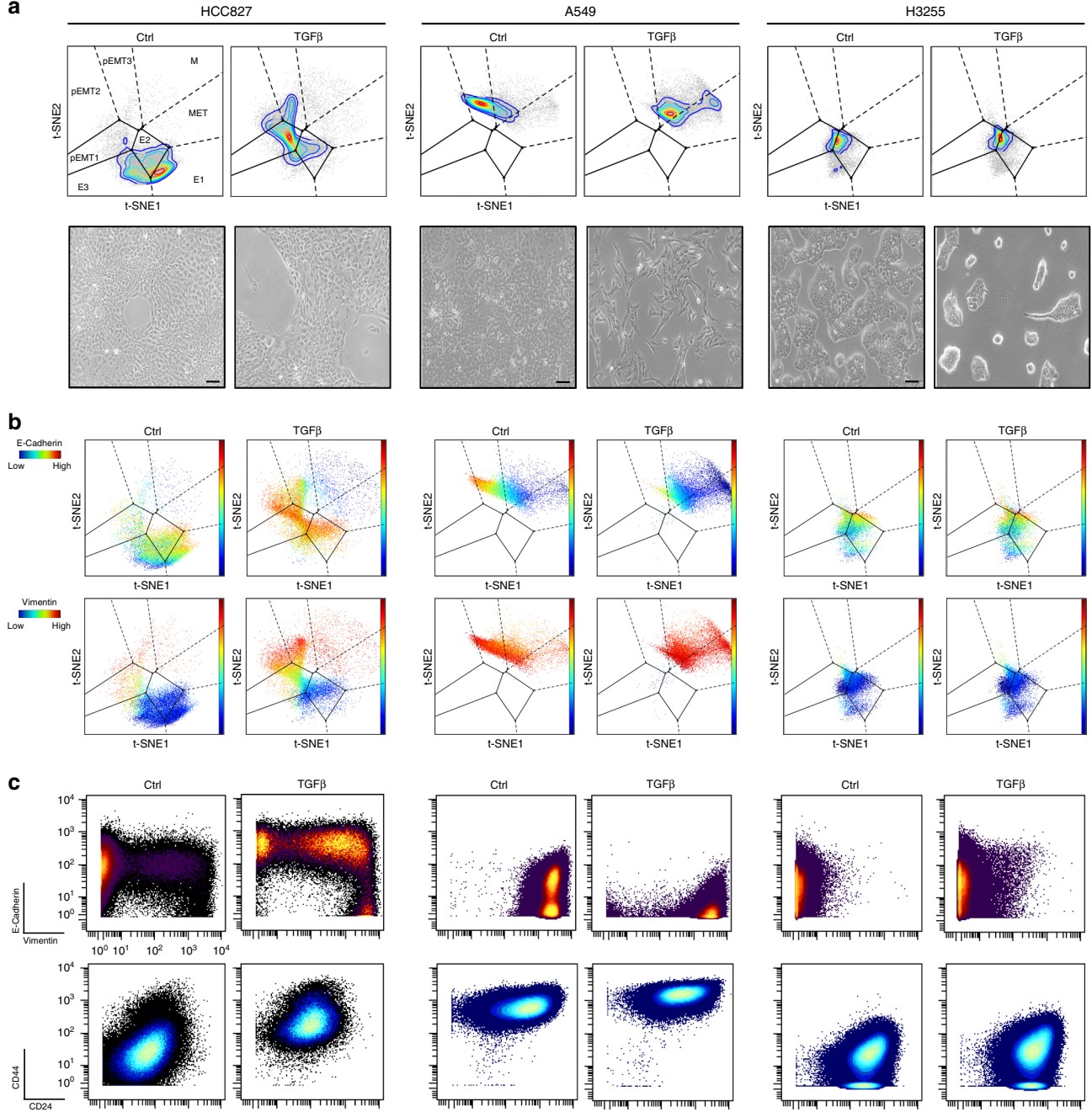

**Fig. 5 EMT–MET PHENOSTAMP serves as a means to assess EMT–MET states in independent NSCLC cell lines. a** Projection of NSCLC cell lines onto the EMT–MET PHENOSTAMP (top) and their respective morphological assessment (×10 images, scale bar 200 μm, bottom). HCC827 samples (ctrl, 2-day TGFβ) were ran on a different day than the ones used for the map construction for validation (left). A549 and H3255 cells were treated with TGFβ for 10 and 13 days, respectively, and analyzed with mass cytometry. **b** Shown are the respective to each cell line and condition expression profiles of E-Cadherin (top) and Vimentin (bottom) visualized on the EMT–MET PHENOSTAMP. **c** E-Cadherin/Vimentin and CD44/CD24 mass cytometry plots per projected cell line sample. See Methods and Supplementary Fig. 8 for further details.

it does not account for the expression of intracellular markers that would offer insights on each state's cell cycle and death kinetics, features that could better inform state transitions. Our data show that M cells express significantly lower levels of the mitotic marker pH3[50] compared to MET cells by ~34% (Supplementary Fig. 7 and Supplementary Table 3), suggesting that our hysteresis findings (M to MET transition) are not due to an expanding population of cells. Future studies involving live cell tracing and incorporation of marker expression in our analysis will be critical towards deciphering EMT–MET dynamics.

We introduce a neural net algorithm for projecting samples on the EMT–MET PHENOSTAMP with single-cell resolution. This map can be used as a potential tool to assess NSCLC clinical specimens in terms of their EMT status, as defined along a well-established and reproducible in vitro time-course analysis. Our mapping of five clinical samples serves as proof of concept that PHENOSTAMP can offer insights on the clinical relevance of of EMT and MET states. For example, we observed a potential association of EMT with immune infiltration. Such a correlation has been previously reported but not linked to mutation status[51],

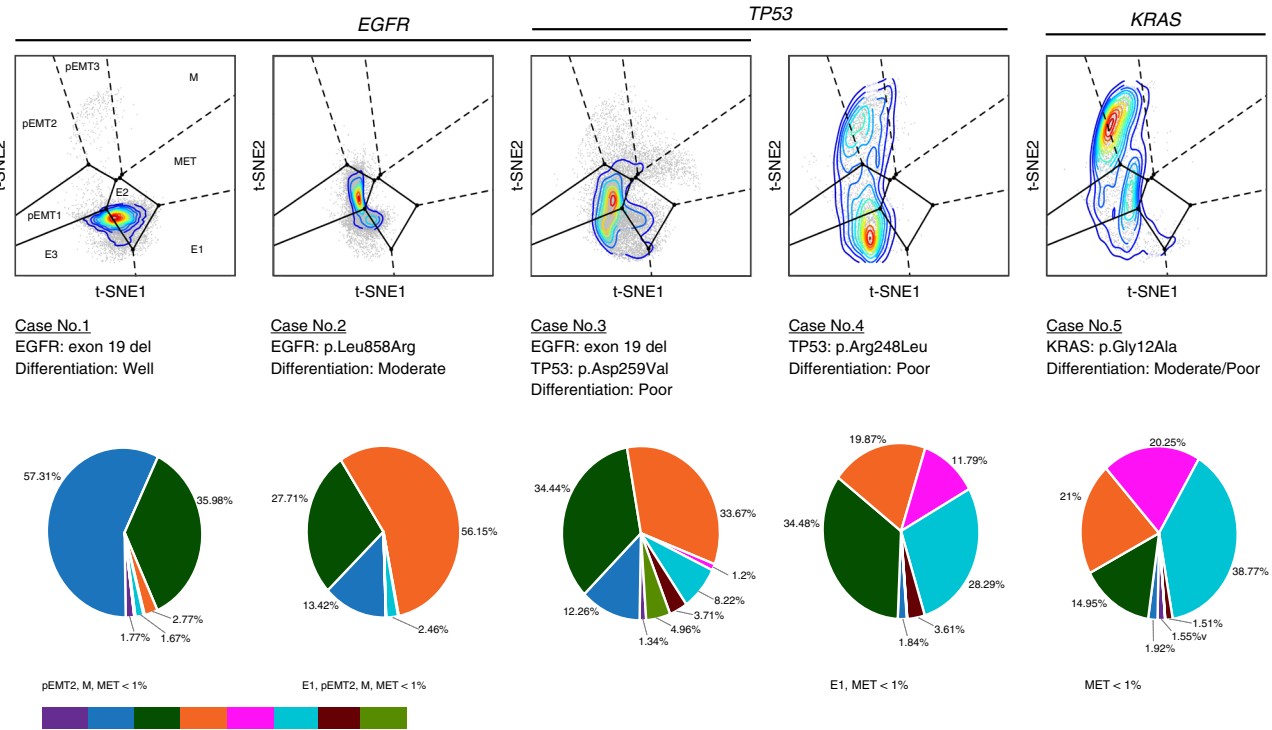

**Fig. 6 EMT–MET PHENOSTAMP demonstrates the existence of partial and full EMT and MET states in NSCLC clinical samples.** Shown are the projections of five clinical samples analyzed with mass cytometry and their respective mutation and differentiation features. Pie charts show % of tumor cells per clinical sample mapping on the color-coded computationally derived EMT–MET states. See Methods, Supplementary Table 4, and Figs. 9 and 10 for further details.

supporting the necessity of a larger scaled study and the importance of a well-defined EMT reference map. Furthermore, PHENOSTAMP can help detect EMT-related drug-resistant or -sensitive cell phenotypes[8,52,53] per patient tumor sample. For instance, studies suggest that inducing MET could be beneficial by restoring sensitivity to TKI inhibitors[16,54]. Indeed, our data showed that pEGFR signaling increased by ~20% in cells undergoing MET compared to cells in the M state, and this would hypothetically constitute TKI sensitivity (Supplementary Table 3). Our data also showed that cells in the MET state significantly differ from M cells in MUC1 expression, a molecule that is being studied as a lung cancer therapeutic target[55]. Therefore, depending on the situation and timing, one may choose to target or induce the MET state. Having an EMT–MET map that can theoretically distinguish whether a specimen has cells undergoing EMT or MET with targetable markers could benefit such future efforts.

Our study proposes an approach for interrogating and mapping EMT states, but has its limitations. Although here we chose TGFβ to induce EMT in lung cancer cells, it is well known that EMT can be triggered by a variety of conditions, including hypoxia and drug treatments[56]. It would be therefore reasonable to compare different modes of EMT induction and examine how these would affect the EMT–MET map, or the projections onto it. In addition, incorporating different mutational backgrounds in our analysis would also be a worthy pursuit. It is important to note that projecting individual specimens on a reference map differs from traditional approaches where multiple samples are pooled for analysis, allowing each specimen to be examined within its own heterogeneous identity. Also, by using single-cell proteomic data, our approach differs from similar studies that have employed basic concepts from machine learning to classify and diagnose clinical samples based on gene profiling alone[36,57].

Functional characterization of each state is also worth pursuing, given that this would inform each state's metastatic, drug resistance and stem-like properties. For example, although we used common markers (CD44hi/CD24lo) for defining CSC-like cells, we did not functionally test stemness of these cells. In this regard, it has been shown that CD44hi/CD24hi pEMT cells can exhibit stem-like behavior[11,58]. Given that we show that pEMT cells are plastic in nature (TRACER results), examining the stemness and drug resistance of the identified pEMT states would be warranted. Finally, our work can be extended to multiplexed imaging[59] to delineate the spatiotemporal relationships between EMT malignant cell phenotypes and composition of the surrounding microenvironment.

In summary, we have defined a landscape of EMT and MET states with single-cell resolution from an in vitro time-course analysis where we modulated EMT states with TGFβ treatment and withdrawal. With this information, we created PHENOSTAMP that could be used in the future to phenotypically assess clinical samples. Furthermore, our experimental and computational approaches provided insights on aspects of EMT basic biology, EMT/MET trajectories, and the heterogeneity of transient cell subpopulations undergoing phenotypic transitions. Combining our work with functional characterization of the states we have presented here and incorporation of single-cell data across more patient samples will enable a more a comprehensive understanding of the relevance of EMT in cancer.

## Methods

**Cell culture**. HCC827, H3255, and A549 NSCLC adenocarcinoma cell lines were a generous gift from Dr. Parag Mallick and were grown in RPMI and DMEM (Dulbecco's modified Eagle's medium) media respectively, supplemented with 10% fetal bovine serum (FBS), and 5% antibiotic solution (penicillin/streptomycin), at 5% $CO_2$ and 37 °C. Cell lines were not tested for mycoplasma contamination or authenticated for this study.

**EMT induction in NSCLC cell lines**. For optimal EMT induction cells were seeded in 100 mm tissue culture plates at 750,000 cells/plate for 24 h and then switched to 2% FBS media for 24 h. Cells were then treated with 5 ng/mL TGFβ (R&D #240-B/CF) for various time-points in 2% FBS media. To keep cell density a non-critical factor for EMT induction, cells were re-seeded (same number of cells/plate) at each collection time-point through the entire course of the treatment. For withdrawal conditions, cells were seeded from TGFβ-treated cells in the absence of ligand and collected at same time increments as with TGFβ treatments with continuous re-seeding as mentioned above. In certain cases, TGFβ was added to cells for a period of 1 or 2 weeks with or without re-seeding as indicated in respective figure legends.

**Human studies**. Clinical aspects of this study were approved by the Stanford IRB in accordance with the Declaration of Helsinki guidelines for the ethical conduct of research. All patients involved provided a written informed consent. Collection and use of human tissue was approved and was in compliance with data protection regulations regarding patient confidentiality and were approved under Stanford IRB protocol #15166. Following surgical resection of primary tumors from five patients at Stanford hospital, NSCLC adenocarcinoma specimens were immediately processed in order to achieve single-cell suspensions for mass cytometry analysis.

**Immunoblotting**. Cells were lysed by adding lysis buffer (50 mM Tris, pH 8.0, 2% SDS, 1× protease inhibitor cocktail, 25 mM NaF, 100uM Na₃VO₄, 5 mM EGTA and 5 mM EDTA) to adherent cells (using a cell scraper). Lysates were then sonicated on ice in order to shear DNA and reduce viscosity. To pellet cellular debris, lysates were centrifuged at $18,000 \times g$ at 4 °C for 10 min. Protein quantification was performed using the Pierce BCA Protein Assay Kit (Thermo Scientific) and equal amounts of total protein were subjected to standard electrophoresis conditions. Separated protein lysates were transferred to PVDF membranes (Millipore #IPVH00010), which were subsequently rinsed in TBST (Tris-buffered saline, 0.1% Tween-20). Blocking was performed using 5% (w/v) nonfat dry milk in TBST for 1 h at room temperature, followed by three 5-min washes in TBST. Membranes were incubated with primary antibody solutions (using 5% milk or BSA in TBST) overnight at 4 °C at the respective dilutions: E-Cadherin (BD, #610181, 1:5000), Vimentin (Abcam, ab92547, 1:500), CD44 (CST, #3570, 1:200), Zeb (CST, #3396, 1:200), Slug (CST, #9585, 1:200), Twist (GeneTex, #GTX127310, 1:500),and GAPDH (CST, #5174, 1:10,000). Membranes were then washed three times in TBST, 5 min each time, and incubated with appropriate secondary antibodies in blocking solution for 1 h at room temperature. Following three 5-min washes with TBST, immunoreactivity was visualized using a standard ECL Detection System. Uncropped/unprocessed scans of all blots shown in the manuscript are provided in the Source Data file.

**Phase-contrast and confocal fluorescence microscopy**. Phase-contrast images of cells undergoing EMT were obtained with a Zeiss Axiovert 40C microscope (×5, ×10, ×20 objectives) equipped with an Axiocam 105 color camera and processed using the Zen 2 software. For confocal fluorescence microscopy, HCC827 cells that were previously grown in either control conditions (absence of TGFβ, 2% FBS RPMI media) or in the presence of TGFβ (5 ng/ml) for a number of days were seeded on coverslips placed in 6-well plates at a density of 125,000 cells per well. Treatment with TGFβ was continued depending on time-course requirements. Following treatment, cells were rinsed with HBSS and subsequently with ice-cold 5% bovine serum albumin/phosphate-buffered saline (BSA/PBS) twice. Cells were then fixed with paraformaldehyde solution (PFA, EMS, #15710) at a final concentration of 4% for 15 min at room temperature, followed by three washes with PBS. Permeabilization was performed with 0.1% Triton X-100 (Sigma-Aldrich) for 10 min at room temperature, followed by three washes with PBS. Blocking was performed with 5% BSA/PBS solution for 30 min at room temperature. Cells were then incubated with fluorophore-conjugated primary antibodies for 1 h in the dark (Alexa 488 Vimentin antibody (BD, #562338) and Alexa 647 E-Cadherin antibody (BD, # 324112)). Following three washes with PBS, cells were mounted in Gold antifade reagent with DAPI (Molecular Probes P36935). Images were obtained and analyzed using the inverted Zeiss LSM 880 laser scanning confocal microscope with Airyscan (×40 objective) and Zen Black software, respectively.

**Flow cytometry**. Following treatments cells were lifted off tissue culture plates using TrypLE (Life technologies, #12605-010). After counting and assessing % viability with Trypan Blue exclusion, $1 \times 10^6$ cell aliquots per condition were fixed by adding PFA at a final concentration of 1.6% for 10 min at room temperature. Cells were then centrifuged at $500 \times g$ for 5 min at 4 °C to pellet cells and remove PFA and washed once with cell staining media (CSM, 0.5% w/v BSA, 0.02% w/v NaN₃ in PBS). Cells were permeabilized with methanol solution for 10 min on ice and optionally stored at −80 °C for long-term storage. After two washes with CSM, master mix of antibodies was added to pelleted cells at a total volume of 100 μL for 30 min in the dark at room temperature. Following two washes with CSM, cells were analyzed using a LSR II.UV. Cell viability was assessed using the LIVE/DEAD Fixable Blue Dead Cell Stain Kit (Life Technologies, #L23105) as per the manufacturer's instructions. Fluorophore-conjugated primary antibodies used: PE/Cy7 E-Cadherin (BioLegend, Clone 67A4, #324115), Alexa 488 Vimentin (BD, Clone RV202, #562338), Pacific Blue CD44 (BioLegend, Clone IM7, #103019), APC/Cy7

CD24 (BioLegend, Clone ML5, #311131), Alexa 647 Twist (Bioss, #bs-2441R). LSR II. UV was used for analysis and graphs shown were created in Cytobank (www.cytobank.org, Cytobank Inc., Menlo Park, CA). Flow cytometry and confocal imaging for this project were performed on instruments in the Stanford Shared FACS and Cell Sciences Imaging Facilities.

**Cell line sample processing for mass cytometry**. Cell line samples were processed as previously described[60]. Briefly, following treatments, cells were lifted off tissue culture plates using TrypLE. Cell count and initial % viability were assessed using Trypan Blue exclusion. To assess cell viability for mass cytometry, cell pellets were briefly incubated in 1 mL PBS containing cisplatin (Sigma-Aldrich #P4394, final concentration 0.5 μM) for 5 min at room temperature. Cisplatin reaction was quenched by adding complete RPMI media (10% FBS) and subsequent centrifugation for 5 min at $500 \times g$. Cell pellets were resuspended in cell culture media ($0.5-1 \times 10^6$ aliquots per condition) and were fixed by adding PFA at a final concentration of 1.6% for 10 min at room temperature. Cells were centrifuged at $500 \times g$ for 5 min at 4 °C to pellet cells and remove PFA and washed once with CSM. Cell pellets were resuspended in CSM and stored at −80 °C until all time-points of the same time course or multiple clinical specimens were collected.

**Tumor dissociation and processing for mass cytometry**. Briefly, fresh specimens were immersed and transferred from Stanford Hospital to the laboratory in MACS Tissue Storage Solution (Miltenyi, #130-095-929). After recording tumor weight, obtaining macroscopic pictures and removing fat and necrotic areas, tumors were cut into pieces of 2–4 mm. Tumor dissociation was performed utilizing the MACS Tumor Tissue Dissociation Kit (Miltenyi, #130-095-929) as per the manufacturer's instructions. Tumor-derived single-cell suspensions were centrifuged at $500 \times g$ for 5 min to pellet tumor cells, which were subsequently resuspended in RPMI and applied on a MACS SmartStrainer (70uM, Miltenyi #130093237) for filtration. Following centrifugation ($500 \times g$, 5 min), red blood cells were removed using the Red Blood Cell Lysis Solution according to the manufacturer's instructions (Miltenyi, #130094183). Tumor cells were then washed and resuspended in RPMI. Labeling dead cells and subsequent processing of tumor single-cell suspensions for mass cytometry analysis was performed as described for cell line samples described above. All samples analyzed were assessed >80% viable using Trypan Blue prior to fixation.

**Mass cytometry antibodies**. Antibodies used for mass cytometry analysis, including respective information on antibody clone, vendor, metal isotope, and staining concentration, are summarized in Table S1. Aside from one antibody purchased from Fluidigm (CD104), all antibodies were conjugated to metal isotopes in-house using the MaxPar Antibody Conjugation Kit (Fluidigm) and titrated to determine optimal staining concentrations.

**Mass-tag barcoding and antibody staining for mass cytometry**. To improve staining consistency, samples were palladium barcoded and pooled for staining as previously described[61]. In short, different combinatorial mixtures of palladium-containing mass-tag barcoding reagents in dimethyl sulfoxide were added to each cell line or tumor sample previously resuspended in PBS-saponin solution and mixed with pipetting. Samples were incubated with barcoding reagents for 15 min at room temperature. Reaction was quenched with the addition of CSM, followed by several washes with CSM prior to pooling all samples together to proceed with staining. Surface and intracellular antigen antibodies were separately added into two master mixes in CSM, filtered through a 0.1 μm filter (Millipore #UFC30VV00) and centrifuged at $1000 \times g$ for 5 min to remove antibody aggregates. For tumor specimen analysis, separate staining cocktails using the same concentrations were prepared with the addition of antibodies towards CD45, FAP, and CD31 for gating out immune, stromal, and endothelial tumor populations, respectively, as previously described[13]. Samples were first incubated with surface antibody master mix (total volume of 100 μL) for 30 min at room temperature. Cell samples were then washed with CSM and permeabilized with methanol for 10 min on ice. Following two washes with CSM, samples were then incubated with the intracellular antibody master mix (total volume of ~100–150 μL) for 30 min at room temperature. Samples were washed twice with CSM and resuspended in PBS containing 1:5000 ¹⁹¹Ir/¹⁹³Ir MaxPar Nucleic Acid Intercalator (Fluidigm) and 1.6% PFA to stain DNA and stored at 4 °C for 1–3 days. Prior to mass cytometry analysis, cells were washed once with CSM, twice with filtered double-distilled water, and finally resuspended (~10⁶ stained cells/mL) in filtered double-distilled water containing normalization beads (EQ Beads, Fluidigm). During event acquisition, pooled filtered cell samples were kept on ice at all times and introduced into the CyTOF 2 (Fluidigm) using the Super Sampler (Victorian Airship and Scientific Apparatus, Alamo, CA, USA). Apart from antibody metal isotopes listed in Table S1, we also recorded event length, barcoding channels (¹⁰²Pd, ¹⁰⁴Pd, ¹⁰⁵Pd, ¹⁰⁶Pd, ¹⁰⁸Pd, ¹¹⁰Pd), normalization beads (¹⁴⁰Ce, ¹⁵¹Eu, ¹⁵³Eu, ¹⁶⁵Ho, ¹⁷⁵Lu), DNA (¹⁹¹Ir and ¹⁹³Ir), and dead cells (¹⁹⁵Pt and ¹⁹⁶Pt).

**Mass cytometry data processing**. Normalization and single-cell debarcoding were performed through respective algorithms as described previously[61] and

transformed using the inverse hyberbolic sine (ArcSinh) function with a cofactor of 552. Debarcoded samples were uploaded as separate FCS files and analyzed on Cytobank. Non-viable (cisplatin-positive) and apoptotic (cleaved PARP and/or cleaved Caspase-3) cells were removed for all subsequent single-cell analysis (Supplementary Fig. 2). Cytobank software was used for traditional cytometry statistics and visualization (histograms, density plots, heatmaps) and SPADE analysis[62]. Briefly, all data presented for NSCLC cell lines and clinical specimens contained 13,000 to 250,000 and 11,000 to 77,000 viable, non-apoptotic single-cell events, respectively, per experimental sample. For SPADE analysis of clinical specimens (Supplementary Fig. 9), downsampled events target was set at 10%, and the number of nodes target was set at 120. Markers CD45, CD31, FAP, and cytokeratins 7 and 8 were used for clustering.

**CCAST clustering**. Raw mass cytometry data from cell line TGFβ time-course experiments were arsinh transformed on Cytobank. After gating out dead and apoptotic cells, remaining cells from all time-points were pooled together. The CCAST-recursive partitioning-based algorithm[25] was applied assuming a minimum of eight clusters on a subsample of ~96,000 cells obtained from density-dependent down sampling[62] and based on six EMT clustering markers (E-Cadherin, Vimentin, CD44, CD24, MUC1, and Twist). These markers were selected using an unbiased non-parametric regression tree analysis[63]. All six markers were among the most statistically significant markers that correlated independently with the top three principal components explaining about 50% variability in the data ($p$ value <0.001, global test of independence constructed by means of the conditional distribution of linear statistics in the permutation test framework[64]) (see Supplementary Table S2). Final selection of the six markers was based on EMT biological relevance. The CCAST analysis resulted initially in 13 clusters (see Supplementary Fig. 6), of which five were excluded from downstream analysis based on a pre-set threshold of each cluster ≥1% of total number of cells analyzed. All data processing and clustering analysis was performed in R.

**FDL visualization of CCAST clusters**. To visualize the spatiotemporal dynamics of EMT and MET processes in HCC827 cells treated with TGFβ, we used Vortex, a graphical tool for visualizing clusters generated from multiparametric datasets. Specifically, we created single-cell FDLs. The resulting graph was constructed specifically on the pre-identified eight CCAST clusters. We utilized Vortex to only implement edge connections between subsequent time-points (similar to the FLOW-MAP algorithm[65]). The FDL graph was used to assess the EMT–MET phenotypic continuum in terms of timing and the expression profiles of all 28 markers measured with mass cytometry. The graph is built by repulsing all cells with forces proportional to how dissimilar they are in multidimensional protein expression space, while edges hold adjacent cells together by constant spring-like forces[28].

**EMT–MET PHENOSTAMP construction**. Using the six CCAST clustering markers (E-Cadherin, Vimentin, Twist, CD24, CD44, and MUC1), we next generated a 2D t-SNE[66] map using the Rtsne package implemented in R with a default perplexity parameter of 30. Next, to define regions of the map most uniquely associated with each state, we determined the highest-density region per state per sampling time and estimated its respective centroid. We next applied Voronoi mapping using eight optimal center points obtained from the above CCAST clusters. These points were obtained from cluster-specific bins with sizes estimated by varying sizes of 1 to 8 units over time and selecting the size showing the largest adjacent change in the number of cells. To place a continuous boundary on the t-SNE map, we next applied a generalized Convex Hull mapping algorithm, which combines a Voronoi diagram and Delaunay triangulation[67] using the alphahull R package. The combined t-SNE-Voronoi mapping-Convex Hull analysis produces a 2D EMT–MET state space partition map capturing the plasticity of EMT and MET processes.

**Neural network for projecting onto the EMT–MET PHENOSTAMP**. To model the non-linear structure of the underlying 2D EMT–MET state map, we constructed a single-layer-hidden artificial neural network model using the above six markers as input and the reduced 2D t-SNE feature space as a bivariate response. Specifically, we used the six input expression values for each cell from the cell line samples to predict the corresponding position of each new cell onto the 2D reference map space. Projection results were visualized as bivariate scatter plots. To train the network, weights on the edges are modified to minimize the error in the output. At the end of training, the inputs which are most important in prediction have the largest weights, while those that are less important have lower weights. The network was trained on 90% min–max normalized data using the "nnet" R package and the remaining 10% was used (1) to optimize for the 11 hidden nodes needed and (2) to carry out a 10-fold cross-validation on the stability of model predictions. The trained neural network was next used to project an independent HCC827 time-course repeat, A549 and H3255 cell line samples, and five clinical specimens. A $k$-nearest neighbor classification on the partition centers was carried out for each sample to estimate the densities of cells in various EMT state partitions on the map.

**TRACER algorithm**. To more rigorously test the hypothesis of hysteresis when comparing EMT and MET, we developed a trajectory reconstruction analysis TRACER that does not rely on pseudotime assumptions and allows branching. We assumed that EMT and MET are classical Markov processes[35] with constant transition probabilities between the states within EMT and MET but can differ between EMT and MET. Because our observations of the proportion of cells in each state are made at discrete, nonuniform time-points, and because we do not observe the trajectories of individual cells, we cannot employ standard methods to estimate for the state transition probabilities[68]. Also, because the number of states is relatively large compared to the number of observations, we employed sparsity assumptions[69]. We modeled the state transition probabilities $p_{jkt}$ between states $j$ and $k$ at time $t$ by imposing sparsity penalties on $p_{jkt}$ for $j$ not equal to $k$, but no penalty on $p_{jjt}$, with the idea that there is no cost for staying in a state, but switching between states is discouraged. Under these assumptions, the transition probabilities EMT and MET can be estimated by convex optimization, with the sparsity parameter $\lambda$ selected through cross-validation. We chose by the 1-standard error rule[70], that is, we chose the largest $\lambda$ such that its error is within 1-standard error of the error of the minimizing. This leads to a more parsimonious solution. Using bootstrap analysis, we provide a distribution of the transition probabilities for EMT and MET, as shown in Fig. 4e. For the bootstrap analysis, we first independently sampled a multinomial distribution for cell counts in each state. We split the bootstrap sample into 2-folds. The first fold was used to estimate the sparsity parameter $\lambda$. We evaluated the estimator with the chosen $\lambda$ on the second fold. For representative EMT and MET networks, the medoid networks for EMT and MET was selected among all the bootstrap samples as the network for which the average distance to all other networks (entry-wise L1 distance between transition matrices) was smallest (Fig. 4f).

**Reporting summary**. Further information on research design is available in the Nature Research Reporting Summary linked to this article.

## Data availability

The mass cytometry time-series data have been deposited in the Cytobank Stanford database under the name "Karacosta et al. HCC827 EMT time-series mass cytometry data" [https://stanford.cytobank.org/cytobank/experiments/26555]. All the other data supporting the findings of this study are available within the article and its Supplementary Information files and from the corresponding author upon reasonable request. A reporting summary for this article is available as a Supplementary Information file. The source data underlying Figs. 1a, e, 3b, c, 4e and Supplementary Fig. 1g are provided as a Source Data file and these include all raw single-cell data resulting from the CCAST analysis described in the paper.

## Code availability

PHENOSTAMP and TRACER codes used for the construction of the EMT–MET PHENOSTAMP and projections and the interrogation of EMT–MET trajectories, respectively, have been deposited on GitHub under [https://github.com/anchangben/PHENOSTAMP] and [https://github.com/nignatiadis/TRACER] and are available to the public. PHENOSTAMP relies on R (≥3.5.0) and other R libraries. The implementation provided on the github link runs on Mac OS X 10.7 and above operating systems. The TRACER algorithm has been implemented as a R package (R version 3.6.1), which uses the tidyverse (version 1.2.1), ggplot2 (version 3.2.1), and JuliaCall[71] (version 0.16.6) packages. The optimization problem is solved in Julia (version 1.2) through the JuMP.jl[72] (version 0.18.6) and Gurobi.jl (version 0.7.2) packages by interfacing to the Gurobi optimizer (version 8.1.0).

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

## Acknowledgements

We gratefully acknowledge Dr. Wendy J. Fantl for discussions and technical advice on analyzing primary clinical specimens with mass cytometry and for a valuable review of our manuscript. We thank Dr. Matt van der Rijn and Sushama Varma for discussions and input on NSCLC histology and pathology, Kelsey Ayers for assisting initial clinical specimen acquisition, Drs. Tyler Risom and Zinaida Good for their input on data presentation, and Trevor Bruce for mass cytometry technical assistance. We thank Dr. Parag Mallick for providing the NSCLC cell lines used in this study. L.G.K is supported by the NIH/NCI training grant R25CA180993 and the Tobacco-Related Disease Research Program T29FT0569. B.A is supported by the Chan Zuckerberg Initiative DAF. S.C.B is supported by the Damon Runyon Cancer Research Foundation Fellowship (DRG-2017-09) and the NIH 1DP2OD022550-01, 1R01AG056287-01, 1R01AG057915-01, 1-R00-GM104148-01, 1U24CA224309-01, and 5U19AI116484-02. S.K.P is supported by the NIH U54CA209971, the NIH/NCI R25CA180993, and the Chan Zuckerberg Initiative DAF.

## Author contributions

Conceptualization: L.G.K., B.A., S.C.B., S.K.P.; methodology: L.G.K., S.C.B., B.A., N.I, R.T., S.K.P.; investigation: L.G.K., S.C.K., S.K.P.; software and formal analysis: L.G.K., B.A., N.I., R.T., S.K.P.; resources: J.A.B, J.B.S, S.C.B., S.K.P.; visualization: L.G.K, B.A., S.K.P.; writing original draft: L.G.K., S.C.B., S.K.P.; writing review and editing: all authors; supervision: S.C.B., S.K.P.; funding acquisition: L.G.K., B.A., S.C.B., S.K.P.

## Competing interests

The authors declare no competing interests.
