## [Peer Review File · Nature Communications]

Reviewers' comments:

Reviewer #1 (Remarks to the Author):

This manuscript addresses important issues pertaining to EMT and the transition from EMT to MET. The study is well designed and is particularly informative as it relies on mass cytometry for assessment of the EMT state in lung cancer cell lines and provides insights at the single cell level. They applied a computational to compare state transitions in EMT and MET. Moreover the authors have utilized a neural network algorithm to phenotype the EMT status of clinical samples at the single cell level.

I do not have issues or concerns with the study design and interpretation of findings. Unfortunately as in most studies of EMT there is little that can be gleaned regarding actionable therapeutic strategies to overcome EMT and its association with drug resistance and poor outcome. Nevertheless overall the study is well designed and the findings are relevant to the EMT field.

Reviewer #2 (Remarks to the Author):

Epithelial-mesenchymal transition (EMT) is an essential evolutionary conserved developmental program that can be reactivated in pathological contexts. As pointed out in this manuscript, the EMT is not a binary process, as transition from the epithelial to the mesenchymal state occurs through a succession of semistable states, partial EMT and partial MET states (for the reverse process). A better characterization of EMT states and how cells can navigate through them is required to understand phenotypic plasticity not only during development but also in cancer.

In this manuscript, Karacosta and colleagues use mass cytometry to map EMT/MET states through time-course analysis of TGFb-induced EMT in cultured lung cancer cells. The authors also use computational tools to map EMT/MET trajectories and apply their EMT/MET road map to patient samples to assess the clinical significance regarding metastatic burden and response to therapy. Although the manuscript tackles a relevant question using elegant technics and powerful computational analyses, the study is purely descriptive and correlative and does not provide useful novel information.

Main concerns

1- The authors use the expression of E-cadherin, Vimentin and the stemness associated markers CD24 and CD44, assessed by FACS (Figure 1) or by mass cytometry (Figure 2). Surprisingly, the gating strategy relies exclusively on CD24/CD44 relative expression while the four gates in Figure1-C are used to define E, pEMT, pEMT/M and M states, respectively. The projection of these gates in Ecadh/Vimentin plots show that gate 1 and 2 are both a mix of E/pEMT in steady state and therefore, the gating strategy cannot predict E and pEMT states in the untreated cells, which are at the starting point of the mapping. Furthermore, the 4th gate (red), depicting a CD24^{low}/CD44^{high} cell population, is described by the authors as the 4th EMT state. When represented in an Ecadh/Vim plot, this M* group fully overlaps with the most mesenchymal cell (low E-cad) of pEMT/M group. There is no proof that the M* population is at an EMT state different from a significant part of the pEMT/M group. Thus, the definition of the states is quite loose and the combination of markers used fails to discriminate between EMT states with enough resolution.

2- In Figure 3, using a multidimensional analysis, the authors retain 6 markers from the mass cytometry single cell analysis. These correspond to the four mentioned above, plus the EMT transcription factor Twist and the epithelial marker Mucin1 (MUC1). They are used to assign

different EMT states in their EMT/MET in vitro model. Interestingly, this analysis segregates the cells all over the time course in 8 groups, but it is not clear how these 8 groups can be assigned to 8 EMT states. The most striking distinction is that between pEMT2 and pEMT3 and between M and pMET. The molecular discrimination is based on the expression of only one epithelial marker, MUC1, which is clearly not sufficient to assign independent EMT states. Similarly, the 8th group, referred to as pMET, can as well be a M variant with lower MUC expression. Sustained TGFb can result in a stable mesenchymal phenotype in a cell-autonomous manner. Unfortunately, the analysis performed cannot discriminate between a partial MET state or a stabilized M state based on the expression of two genes.

3- The reconstruction strategies, such as TRACER, are powerful tools that rely on the transition probabilities between states. Evaluating this approach is beyond our expertise, but we have one comment that needs to be clarified. The cells used in this analysis are highly heterogeneous with respect to the E-cad/Vim ratio (Figure 1-C Ctrl). This heterogeneity is reflected by the distribution of the untreated cells within three E groups (Figure 4C). These groups are not equally represented in steady states and they can very likely respond to TGFb with different kinetics in their transition towards the pEMT or M states, and they can follow similar or different trajectories. While these results uncover an unprecedented level of complexity, it is very difficult to conclude that the transition from E to M is following one trajectory as claimed by the authors.

4- The authors assume that the change in the distribution of the cell population results mainly from the transition between the different states, but do these groups display similar cell proliferation rates, cell-cell adhesion or cell death during TGFb administration or TGFb withdrawal? These parameters need to be taken into account as they likely have a significant impact into the final size of each subpopulation at a given time point. In other words, how does this analysis take into account the heterogeneity in cell behavior?

Reviewer #3 (Remarks to the Author):

In this manuscript by Karacosta et al., the authors have used mass cytometry to describe the range of phenotypes of lung carcinoma tumor cells as they in relation to the process of epithelial-mesenchymal transition. Focusing on one of three different cell lines evaluated, the authors have analyzed the evolution of cellular phenotypes induced by 10 days of stimulation with TGF-beta followed by withdrawal of TGF-beta to also profile the reversion of phenotypes. Several novel methods are used describe the phenotypic trajectory of these cells and in a way that can be applied to subsequently collected data. This is then applied to the analysis of primary lung cancer cells to show heterogeneity and some variation in the extent to which these tumor cells display characteristics of mesenchymal transition.

The paper is nicely written, and the data appear to be high quality (mass cytometry data nicely compared with fluorescence flow cytometry data and the authors have provided examples of raw data is appreciated). The analysis is also rigorous, thoughtful, interesting, and this topic is important. Although the novel methods are interesting and appear to have merits, I am concerned that these new methods might overly complicate the overall message of the paper. It is also not clear why these methods are needed when there are already a range of similar methods available that might be able to achieve the same purpose in a manner that might be easier for a typical reader to digest. Nonetheless, I think this paper is interesting and important.

Specific comments:

How does TRACER compare to other trajectory mapping methods? Why was it important to develop this method for this project? Minor: given that this project is related to cancer, I wonder if

the authors should be concerned about the similarity of this name with TRACERx, an ongoing research project on the study of cancer evolution.

It is appreciated that several marker raw data shown. Is this low staining expected for slug and snail? Was staining for these confirmed with other cells. About pSMAD2/3 staining, these plots make it impossible to see the baseline data. Is it correct to say that pS6 was negative in the M* state given the high staining shown in Fig. 2. Perhaps change to relatively lower staining? What is the basis for choosing the CCAST clustering algorithm. This does appear to be an interesting method but it is insufficiently introduced in the introduction. Some brief description of what this does would be appreciated.

It seems like the gating strategies produced by this algorithm could have been used in place of the STAMP method that the authors have developed. What is important about the STAMP method that other metaclustering approaches? Did the authors simply try using the gating strategies defined by CCAST on each of the samples to assess the relative contributions of each of these clusters?

Related to Supplementary Table 2, please show principal component loadings in addition to these p values.

Although the force directed layout plots are very nice. This method introduces bias into the visualization by incorporating information about sample order. Did the authors compare to other analyses that are typically used for visualizing trajectories such as diffusion map or UMAP?

The last sentence of the abstract is an overstatement given the small number of samples analyzed and the lack of any statistical analysis this should be modified to be more accurate.

About software availability, given that there are a number of less standard and novel analysis tools used here and in various combinations, I think that software or code used to generate the data should be made available.

Reviewer #4 (Remarks to the Author):

In general I think this is a very worthwhile study. First, there has been a relative paucity of time-resolved studies of EMT and (especially) MET; the typical data in the field have only two or three time points. Second, the emphasis on single-cell measurements is critical if one wants to disentangle any process that is likely to involve heterogeneous populations of cells. In general I found the data quite believable (and I was happy with the attention to such details as the effects of confluency) and the analysis methods quite informative. Thus, in my opinion this work deserves publication.

Of course I did have some requests for clarification of various points that I found unclear. First, the repeated use of the word "continuum": when talking about the state space was confusing. The authors find a finite number of states (eight for their detailed mass-spec data with "minimal within-state marker heterogeneity" (p.8)). So, why do they conclude that there is no useful way to identify discrete states and that the states are best thought of as forming a continuum? This is a topical question as most recent studies argue in favor of some relatively small number of major phenotypes (see Patsushenko et al, *Nature* 556 (7702), 463 (2018); Jia, Dongya, et al. "Testing the gene expression classification of the EMT spectrum." *Physical biology* (2019)) whereas some modeling studies suggest something like a continuum (Font-Clos, Francesc, Stefano Zapperi, and Caterina AM La Porta. "Topography of epithelial-mesenchymal plasticity." *Proceedings of the National Academy of Sciences* 115.23 (2018): 5902-5907) - perhaps this issue should be discussed in greater detail and some of these references added.

Another point where the discussion is perhaps over-simplified is the initial statements and subsequent validation of the STAMP approach using for example A549 cells. As far as I can tell the A549 data was not single cell and therefore one cannot conclude that all the cells are in a pEMT state. In fact previous studies of the EMT spectrum of A549 cells (George, Jason T., et al. "Survival outcomes in cancer patients predicted by a partial EMT gene expression scoring metric." *Cancer research* 77.22 (2017): 6415-6428 - see Figure 3 therein) argue that A549 is rather inhomogeneous, with some pEMT cells but many E and M as well; averaging this over the population by making bulk measurements is rather misleading. I was very happy to see that the comparison with the patient samples was done at the single cell level and does not suffer from this confusion.

The authors take as given the somewhat standard view that stem cells exhibit a Cd24-CD44+ signature. This appears to be correct for mesenchymal stem cells, but recent work in the breast cancer context (Grosse-Wilde, Anne, et al. "Stemness of the hybrid epithelial/mesenchymal state in breast cancer and its association with poor survival." *PLoS one* 10.5 (2015): e0126522; Goldman, Aaron, et al. "Temporally sequenced anticancer drugs overcome adaptive resistance by targeting a vulnerable chemotherapy-induced phenotypic transition." *Nature communications* 6 (2015): 6139. has suggested that there can one stem-like behavior associated with pEMT and with CD24+CD44+ markings.

The results about hysteric features of the E->M->E loop and the new type of pEMT state are quite important and as argued tend to agree with previous arguments, albeit with a lot better data here. It would be worth highlighting a bit more the idea that is partially mentioned here that the return trajectory might depend on exactly how long the TGF-beta stimulation was applied for. Here it seems that applying it for less time makes return (mostly from pEMT states) more reversible, but the same might be true for much longer stimulation protocols. This may be related to the percentage of cells that do not return to E under this protocol. This idea may eventually merge into results for MET on sarcoma cell lines (which have been M for a very long time) - see Somarelli, Jason A., et al. "Mesenchymal-epithelial transition in sarcomas is controlled by the combinatorial expression of MicroRNA 200s and GRHL2." *Molecular and cellular biology* 36.19 (2016): 2503-2513.

Unless I am missing something, there appears to be some confusion in Figure numbers on p 6 (lines 120-125). Could the authors check these?

Herbert Levine

*

The authors would like to sincerely thank the reviewers and the editor for their time and providing us with their thoughtful and beneficial critiques of our manuscript. **Changes from previous manuscript version highlighted in revised manuscript text.**

After thoroughly studying the reviewers' comments, the following point-by-point responses have been prepared:

Reviewer #1 (Remarks to the Author):

Comment:

This manuscript addresses important issues pertaining to EMT and the transition from EMT to MET. The study is well designed and is particularly informative as it relies on mass cytometry for assessment of the EMT state in lung cancer cell lines and provides insights at the single cell level. They applied a computational to compare state transitions in EMT and MET. Moreover the authors have utilized a neural network algorithm to phenotype the EMT status of clinical samples at the single cell level.

I do not have issues or concerns with the study design and interpretation of findings. Unfortunately as in most studies of EMT there is little that can be gleaned regarding actionable therapeutic strategies to overcome EMT and its association with drug resistance and poor outcome. Nevertheless overall the study is well designed and the findings are relevant to the EMT field.

Response:

We thank the reviewer for their accurate description of our study on mapping EMT-MET states and trajectories in lung cancer at the single-cell level. We appreciate that the reviewer deems our study as well designed, informative and relevant to the EMT field. We agree, that most studies do not offer actionable therapeutic strategies to overcome EMT and its association with drug resistance and poor outcome. We do however demonstrate parallels with cell states we observe in vitro with primary human lung cancer. We hope that by phenotyping clinical samples in terms of EMT and MET dynamic state changes with single-cell resolution, our approach will eventually link EMT-informed treatment strategies in a clinical context. In ongoing work, we are applying our EMT-MET state map to assess state-specific drug sensitivity traits with the ultimate goal of tackling EMT-related drug resistance in lung cancer in a patient-specific manner.

Reviewer #2 (Remarks to the Author):

Comment:

Epithelial-mesenchymal transition (EMT) is an essential evolutionary conserved developmental program that can be reactivated in pathological contexts. As pointed out in this manuscript, the EMT is not a binary process, as transition from the epithelial to the mesenchymal state occurs through a succession of semistable states, partial EMT and partial MET states (for the reverse process). A better characterization of EMT states and how cells can navigate through them is required to understand phenotypic plasticity not only during development but also in cancer.

In this manuscript, Karacosta and colleagues use mass cytometry to map EMT/MET states through time-course analysis of TGFb-induced EMT in cultured lung cancer cells. The authors also use computational tools to map EMT/MET trajectories and apply their EMT/MET road map to patient samples to assess the clinical significance regarding metastatic burden and response to therapy. Although the manuscript tackles a relevant question using elegant technics and powerful computational analyses, the study is purely descriptive and correlative and does not provide useful novel information.

Response:

We appreciate that the reviewer recognizes the novelty of our analysis and for providing detailed comments, concerns and suggestions to significantly improve the manuscript. However, we respectfully disagree that our work is purely descriptive and correlative. Although part of our work is descriptive, we have also provided

novel computational approaches to quantify EMT and MET states based on dynamic in-vitro data and quantify likely transition probabilities between the states and also provide confidence intervals on our estimates. To the best of our knowledge, we are not aware of studies that perform single-cell analysis of time-dependence that explicitly infers causality. We believe that this distinguishes our work from purely descriptive or correlative studies. Next, we address the relevance of our work. We have provided a new approach to phenotyping clinical samples in context of the EMT spectrum we characterized in our in-vitro systems. In particular, we developed a neural net algorithm to perform relative assessment of EMT-MET states in clinical specimens. This phenotyping provides a novel approach of characterizing clinical samples with single-cell resolution. For example, it demonstrates the presence of signaling states that are likely to be resistant to EGFR-targeted therapy in EGFR+ tumors. Overall, we believe that it is our systematic single-cell dissection of this process, recapitulating years of fragmented observations, combined with parallel annotations of these newly defined states in clinical specimens that represents the novel contribution here. These insights not only validate much of the current knowledge in the EMT field, but also provide a new guide to bridge in vitro findings with translational studies that aim to establish the role of EMT in drug resistance and poor outcome. We provide point-by-point responses to Reviewer #2's comments and suggestions below and have made related changes to our text, references and data analysis/presentation to improve readability and clarity of our manuscript.

Main concerns

Comment:

The authors use the expression of E-cadherin, Vimentin and the stemness associated markers CD24 and CD44, assessed by FACS (Figure 1) or by mass cytometry (Figure 2). Surprisingly, the gating strategy relies exclusively on CD24/CD44 relative expression while the four gates in Figure1-C are used to define E, pEMT, pEMT/M and M states, respectively. The projection of these gates in Ecadh/Vimentin plots show that gate 1 and 2 are both a mix of E/pEMT in steady state and therefore, the gating strategy cannot predict E and pEMT states in the untreated cells, which are at the starting point of the mapping.

Response:

We agree that using only E-Cadherin, Vimentin, CD44 and CD24 biaxial gating strategies provides only a loose definition of EMT states and cannot predict E and pEMT states in untreated cells (which are indeed heterogeneous in marker expression). In our system, CD44 and CD24 yielded more distinct populations of cells to manually gate, of which we then interrogated their expression towards E-Cadherin and Vimentin (conventional markers to assess EMT status). The goal with Figure 1 was to present a typical EMT characterization with flow cytometry (specifically with these 4 markers) and how (as Reviewer 2 points out) misleading this can be. This then serves as a backdrop and justification for deeper single-cell analysis of this process, and thus novel contribution of our findings and refined state characterization here.

The data in Figure 2 simply demonstrate that we can achieve similar analysis by mass cytometry as with flow cytometry – again, grounding our new observations in previously established metrics of EMT. The heterogeneity of E and pEMT states is addressed in Figure 3 of the revised manuscript where the CCAST analysis shows that at least by using our selected clustering markers, we actually observe 3 E states (E1, E2, E3) and 3 pEMT states (pEMT1, pEMT2, pEMT3) and at steady state time “0”, some of these states co-exist to a certain degree, in varying abundances.

We have adjusted the text in the revised manuscript to emphasize these clarifications.

Comment:

Furthermore, the 4th gate (red), depicting a CD24^{low}/CD44^{high} cell population, is described by the authors as the 4th EMT state. When represented in an Ecadh/Vim plot, this M* group fully overlaps with the most mesenchymal cell (low E-cah) of pEMT/M group. There is no proof that the M* population is at an EMT state different from a significant part of the pEMT/M group. Thus, the definition of the states is quite loose and the combination of markers used fails to discriminate between EMT states with enough resolution.

Response:

As mentioned above, the goal in Figure 1 was to present a typical EMT characterization with flow cytometry by specifically using E-Cadherin, Vimentin, CD44 and CD24 expression profiles. As with any approach that aims to define phenotypic states, the derived states are a function of the markers measured and the decisions on how much to subdivide states. There will always be within-state heterogeneity, such that the within-state heterogeneity should be lower than between-state heterogeneity. Hence, all the states, including the M state, have some level of within-state heterogeneity. The M* cells (as pointed out by the reviewer), do overlap with the most mesenchymal cells in terms of their E-Cadherin/Vimentin expression profile, however they are distinctly characterized by high CD44 and low CD24 expression which has been previously reported as being a mesenchymal stem-like phenotype¹. Given the interest and studies on EMT-related stem-like phenotypes, we wanted to highlight this particular observation. Notably, M* cells also exhibit the lowest expression for pS6 and pEGFR compared to the other mesenchymal cells (Supplementary Figures 2F and 5B in revised manuscript). When we performed in depth single-cell analysis, our algorithm did not define M* cells as one of the 8 distinct EMT-MET states, however, mass cytometry analysis reproduced our initial findings that there is a subpopulation of M* cells within the M state with the distinct CD44^{hi} CD24^{lo} pEGFR^{lo} pS6^{lo} phenotype.

While we agree that “state” could be misleading for reasons highlighted by the reviewer, the term (defined by protein markers) is used in studies with all the accompanying caveats². Alternatives like ‘population’ could be used instead, however it would still bare the same caveats in terms of underlying heterogeneity.

To address the concerns regarding definition of the EMT states we have added commentary in the results section pertained to Figures 1-3 of the revised manuscript to better explain our analysis strategy and state definitions, and to improve readability and clarity.

Comment:

In Figure 3, using a multidimensional analysis, the authors retain 6 markers from the mass cytometry single cell analysis. These correspond to the four mentioned above, plus the EMT transcription factor Twist and the epithelial marker Mucin1 (MUC1). They are used to assign different EMT states in their EMT/MET in vitro model. Interestingly, this analysis segregates the cells all over the time course in 8 groups, but it is not clear how these 8 groups can be assigned to 8 EMT states.

Response:

The computationally-defined 8 EMT-MET states resulted from an analysis of our time-series data using our CCAST algorithm³. Briefly stated, CCAST is an algorithm that groups cells by identifying gating/clustering markers, creating a clustering hierarchy whose partitions are optimized to isolate homogeneous subpopulations of cells (addressing the issues raised in the earlier comments). Most importantly, it utilizes a well-established unbiased variable selection and non-parametric statistical technique, based on recursive partitioning to classify members of certain populations of cells in the form of a decision tree. Supplementary Figure 5 of the revised manuscript, contains details on the CCAST decision tree, specifically showing how each marker was used in each step to separate the cells into eventually 13 clusters of cells of which we retained the 8 most prominent ones. For this analysis, our cutoff for retaining a cluster –i.e. EMT state– was number of cells in cluster representing $\geq 1\%$ of the total number of pooled cells analyzed. We could have continued, and break the decision tree even further, using additional markers, ultimately partitioning our map to more “states”, but we wanted to maintain some parsimony for interpretation purposes.

The assignment of these CCAST clusters to the 8 EMT-MET states was performed in mainly 2 ways. First, based on the expression profiles of the widely accepted phenotypic markers E-Cadherin, CD44 and Vimentin (e.g. E-Cadherin and Vimentin expression is inversely correlated in Epithelial and Mesenchymal states respectively and are co-expressed in partial EMT states. CD44 is expected to be low in Epithelial cells, and high in Mesenchymal cells^{1,4}). The annotation of our 8 EMT-MET states is largely in agreement with this. In addition, we leveraged the highly multiplexed nature of our analysis and the expression of additional phenotypic markers in further support of our state assignments. For example, in states E1-E3, not all have high E-Cadherin (e.g. E3 is low in E-Cadherin expression), but the same state is high for other epithelial markers such as TROP2 and Cytokeratin 7 (Supplementary Figure 6 in revised manuscript).

State assignment was also supported by each cluster's prominence as a function of time in our dynamic experiment (Figure 3B, revised manuscript). For example, the majority of cells at time 0 should be in Epithelial states, and number of cells assigned to states E1-E3 should decrease in presence of TGF β . The reverse should be the case for Mesenchymal states. A mesenchymal-epithelial **MET** state (which we initially named "pMET", and have changed to **MET** in the revised manuscript), should be most prominent once TGF β is removed, and should show increase of epithelial features (in our case MUC1). Again, these anticipated patterns were supported by our time-course data.

Accordingly, we have added commentary in the results section of the revised manuscript to better justify our EMT state assignments as outlined here.

Comment:

The most striking distinction is that between pEMT2 and pEMT3 and between M and pMET. The molecular discrimination is based on the expression of only one epithelial marker, MUC1, which is clearly not sufficient to assign independent EMT states. Similarly, the 8th group, referred to as pMET, can as well be a M variant with lower MUC expression. Sustained TGF β can result in a stable mesenchymal phenotype in a cell-autonomous manner. Unfortunately, the analysis performed cannot discriminate between a partial MET state or a stabilized M state based on the expression of two genes.

Response:

As described above, the discrimination of the states was done utilizing the CCAST decision tree. In a first unsupervised step, we used PCA analysis to identify the top statistically significant markers (Supplementary Table 2). From these markers we chose the 6 clustering markers discussed in our manuscript (E-Cadherin, Vimentin, CD44, CD24, Twist and MUC1). It is true that from our CCAST analysis, MUC1 is the marker most significantly different between states pEMT2, pEMT3 and M, MET. MUC1 is one of the most interesting findings of our study, given that not much has been reported on its role in EMT. Although we do not fully understand its significance in mesenchymal-epithelial transition (MET) in cancer, we do know that it is an epithelial marker that is being targeted in lung cancer⁵. Interestingly, MUC1 has been reported to be involved in normal nephrogenesis during MET⁶, which is in agreement with our respective MET findings in lung cancer cells during TGF β withdrawal.

Nevertheless, we agree that using one marker is probably not sufficient to assign independent EMT states. However, these states are significantly different in the expression of more than one marker. We performed statistical analysis comparing all the markers between states pEMT2, pEMT3 and states M, MET and we have incorporated these results in Supplementary Table 3 in the revised manuscript and also below for the reviewer (markers ranked by p-value). This analysis shows that for states M and MET, the most highly ranked markers (in terms of fold change and p value) are TROP2 and Cytokeratin 8 (both epithelial markers), which are significantly increased (~60 and 50% respectively) in the MET state, signifying the switch of cells undergoing MET. Cytokeratin 8 is also highly ranked as being significantly different between states pEMT2 and pEMT3, showing a 70% decrease in pEMT3 compared to pEMT2, corroborating the transition of pEMT to M cells. Furthermore, in data we don't show here, when we remove from our analysis the 6 clustering markers, and perform CCAST among our 8 states, we found that states M and MET are still separated as different cell populations based on the difference in the expression of TROP2.

In regards to the distinction between M and what was previously named pMET, we acknowledge that the use of the name "pMET" is confusing and as mentioned above, we changed instead to "MET". Although we cannot exclude the possibility that the MET state may very well be a variant of the M state, and that a stable mesenchymal phenotype can arise from sustained TGF β treatment, we believe defining it as a mesenchymal-epithelial (MET) state is more appropriate given that 1) it is most prominent in numbers after we remove the EMT inducer, TGF β , 2) MET cells express significantly higher MUC1, TROP2 and Cytokeratin 8 levels than M cells (all considered epithelial markers^{4,7}) and 3) the number of MET cells decrease with time during TGF β withdrawal (although we acknowledge that these cells are not completely gone at 10 days withdrawal). The fact that some MET cells remain after 10 days of withdrawal may suggest that they are in fact stabilized, or that whatever M cells still remain, are giving rise to more MET cells.

pEMT3 vs. pEMT2						MET vs. M					
	logFoldChange	Statistic	Parameter	p-value	Adj p-value		logFoldChange	Statistic	Parameter	p-value	Adj p-value
Cytokeratin 8	-0.733921038	39.46178	15847.42	0	0	TROP2	0.63519163	-42.53892	19202.7	0	0
Nanog	-0.544376294	40.8149	15063.75	0	0	Cytokeratin 8	0.46141363	-24.83361	17692.38	7.56E-134	2.34E-132
pSrc	-0.607557729	35.34402	17038.77	3.90E-264	1.21E-262	Nanog	0.29493824	-23.04585	17203.75	1.31E-118	4.06E-117
EGFR	-0.513034163	29.96899	16083.14	4.56E-192	1.41E-190	pS6	0.26344908	-23.33111	17714.73	9.12E-116	2.83E-114
pS6	-0.311869527	27.57169	15721.81	1.83E-163	5.68E-162	pH3	0.34189283	-20.27695	17209.39	2.33E-90	7.23E-89
pEGFR	-0.474249046	27.54994	16006.93	2.79E-163	8.64E-162	pSrc	0.28489028	-18.71282	16187.7	2.54E-77	7.89E-76
c-Caspase 3	-0.381844616	25.62544	17007.47	3.91E-142	1.21E-140	c-Caspase 3	0.21345468	-15.27348	16260.63	1.30E-54	4.03E-53
pSmad2/3	-0.429967683	25.33125	16226.74	7.13E-139	2.21E-137	pNFKB	0.28484486	-15.61989	16657.43	2.65E-52	8.21E-51
pNFKB	-0.496206675	24.42201	16535.8	1.94E-129	6.02E-128	b-catenin(non-p)	0.21076742	-13.2264	15771.97	1.01E-39	3.12E-38
b-catenin(non-p)	-0.466889161	22.64003	16750.46	8.29E-112	2.57E-110	pEGFR	0.20388849	-13.20525	17158.95	1.28E-39	3.96E-38
pRb	-0.278385205	22.12974	15619.85	7.13E-107	2.21E-105	Oct3/4	0.16019264	-8.606327	16799.11	8.20E-18	2.54E-16
pH3	-0.443815209	21.93679	16095.98	4.00E-105	1.24E-103	pSMAD2/3	0.13855097	-8.301525	16363.19	1.11E-16	3.43E-15
TROP2	-0.389283364	20.73106	15973.29	3.15E-94	9.77E-93	PD-L1	-0.08843838	6.818008	17442.74	9.53E-12	2.96E-10
Oct3/4	-0.329863718	15.88825	15884.16	2.08E-56	6.44E-55	pAMPK	0.13018351	-6.018404	16542.6	1.80E-09	5.58E-08
CD104	-0.220718256	15.37087	16854.42	5.87E-53	1.82E-51	RUNX1	0.07195712	-5.128097	16138.89	2.96E-07	9.18E-06
Notch3	-0.200148538	15.02184	17064.25	1.11E-50	3.45E-49	pRb	0.0454928	-3.660522	16386.71	0.000252	0.007827
pAMPK	-0.285408209	12.36956	16765.75	5.43E-35	1.68E-33	EGFR	0.05858275	-3.555853	16816.33	0.000378	0.011711
Cytokeratin 7	-0.195629592	11.53101	17076.88	1.20E-30	3.71E-29	Snail	0.03501546	-2.702721	16499.43	0.006884	0.213419
PD-L1	-0.140508918	10.78693	16309.43	4.90E-27	1.52E-25	Cytokeratin 7	-0.0553096	2.667408	16368.39	0.007651	0.237194
RUNX1	-0.153164569	10.14558	16976.81	4.06E-24	1.26E-22	Slug	0.01389564	-1.227157	16363.12	0.219781	1
Snail	-0.118568511	8.344606	16909.59	7.69E-17	2.38E-15	Notch3	0.01144468	-1.031609	16579.16	0.30227	1
Slug	-0.079357778	7.27218	16905.7	3.69E-13	1.14E-11	CD104	0.01225016	-0.879444	16692.62	0.379173	1

Statistical Analysis (ttest) performed between pEMT2–pEMT3 and M-MET states respectively. Markers are ranked according to their p-value. The 6 clustering markers used in CCAST (E-Cadherin, Vimentin, CD44, CD24, MUC1 and Twist) were not incorporated for this analysis. Statistics: t-statistics used for calculating the p-values, Parameter: degree of freedom which is the only parameter for the t-distribution similar to mean and variance for the normal distribution, Adjusted p-value: p-values based on Bonferoni corrections

Regarding our assignment to the 8 EMT-MET states we have added text in the results section to better explain our strategy and to improve clarity towards the conclusions drawn. We have also changed the name “pMET” to MET and provide Supplementary Table 3 that shows additional protein expression differences between M and MET states.

Comment:

The reconstruction strategies, such as TRACER, are powerful tools that rely on the transition probabilities between states. Evaluating this approach is beyond our expertise, but we have one comment that needs to be clarified. The cells used in this analysis are highly heterogeneous with respect to the E-cad/Vim ratio (Figure1-C Ctrl). This heterogeneity is reflected by the distribution of the untreated cells within three E groups (Figure 4C). These groups are not equally represented in steady states and they can very likely respond to TGFb with different kinetics in their transition towards the pEMT or M states, and they can follow similar or different trajectories. While these results uncover an unprecedented level of complexity, it is very difficult to conclude that the transition from E to M is following one trajectory as claimed by the authors.

Response:

We thank the reviewer for bringing this to our attention. We agree that the various identified Epithelial states may respond to TGFβ with different kinetics, making it difficult to conclude that there exists only one trajectory from E to M. While it is well established that deep phenotypic trajectory analyses often recapitulate true biological differentiation trajectories^{8,9} it was not an intended interpretation here. In fact, we infer more than one potential trajectories in Figure 4F where we present the various estimated transitional probabilities of which the stronger ones are depicted by more heavily weighted arrows, and these represent more than one possible trajectories that cells may follow from E to M. To clarify this, we have added commentary on the various trajectories cells may follow during EMT, in the results and discussion sections. In addition, we have changed the description of Figure 4D diagram - which we acknowledge as being misleading - showing only one EMT trajectory. Instead of calling it a “schematic hypothetical model”, we name it instead “Conceptual model of trajectories”, of which its main purpose was to show how cells seem to move in the density plots shown in Figure 4C and highlight the perceived/possible differences between EMT and MET trajectories. It is now described in the respective figure legend in the revised manuscript as: 4D) “Conceptual model of density plot-inferred EMT and MET trajectories of transitioning cells...”

Comment:

The authors assume that the change in the distribution of the cell population results mainly from the transition between the different states, but do these groups display similar cell proliferation rates, cell-cell adhesion or cell death during TGFb administration or TGFb withdrawal? These parameters need to be taken into account as they

likely have a significant impact into the final size of each subpopulation at a given time point. In other words, how does this analysis take into account the heterogeneity in cell behavior?

Response:

We agree with the reviewer’s comments that more parameters need to be taken into account in order to have a better assessment of the changes in cell population distributions during EMT with time. Accounting for these additional factors increases the model’s complexity. We are actively working on this problem as it involves integrating ODE-type modeling with state transition modeling, which is largely unsolved in this context. We have added in the discussion a section describing the limitations of our current trajectory analysis and discuss efforts towards incorporating in our future studies not only number of cells per state per time point, but also the expression of intracellular markers that may be associated with for example proliferation rates, cell death etc., that may dictate kinetics and distributions of the various states during EMT with time.

We also provide here and in our revised manuscript (Supplementary Figure 6A subplots and Supplementary Table 3) single-cell data related to state-specific cell proliferation measurements. Specifically, we show below the expression levels of proliferation markers (pRb and pH3¹⁰) in our 8 EMT-MET states, that may suggest that at least in part, the distributions of cell populations we observe are linked to cell state transitions rather than cell population expansion. In particular, these markers – although they present some within-state heterogeneity - appear lower at what would be considered the 2 ends (i.e. start and end) of the EMT trajectory (E states and M cells – specifically E3 cells and the M* subpopulation of cells described earlier – see circled area on CD44/CD24 t-SNE plots in Figure 4B) compared to states in the middle of the trajectory (states pEMT1-3). Notably, pH3 is in the top 5 statistically significant differentially expressed markers between M and MET states (M being rather low in pH3 overall (line graph below) and ~30% lower compared to MET state (table below)), suggesting that the transition from M to MET (our hysteresis findings) is unlikely due to increased mitotic activity¹¹ of M cells. Nevertheless, as mentioned above, more parameters need to be taken into account in order to have a better assessment of the changes in cell population distributions during EMT with time, and although this was beyond the scope of our current study, we are actively working towards designing better approaches to tackle this important question

(A) Expression profiles of pRb and pH3 markers in pooled HCC827 time point data visualized on t-SNE (EMT-MET state map). (B) To the left, pH3 mean levels across the 8 EMT-MET states in 2 independent biological experiments. To the right, part of table presented in an earlier response showing the statistical analysis results performed between M and MET states. Markers are ranked according to their p-value.

Reviewer #3 (Remarks to the Author):

Comment:

In this manuscript by Karacosta et al., the authors have used mass cytometry to describe the range of phenotypes of lung carcinoma tumor cells as they in relation to the process of epithelial-mesenchymal transition. Focusing on one of three different cell lines evaluated, the authors have analyzed the evolution of cellular phenotypes induced by 10 days of stimulation with TGF-beta followed by withdrawal of TGF-beta to also profile the reversion of phenotypes. Several novel methods are used describe the phenotypic trajectory of these cells and in a way that can be applied to subsequently collected data. This is then applied to the analysis of primary lung cancer cells to show heterogeneity and some variation in the extent to which these tumor cells display characteristics of mesenchymal transition. The paper is nicely written, and the data appear to be high quality (mass cytometry data nicely compared with fluorescence flow cytometry data and the authors have provided examples of raw data is appreciated). The analysis is also rigorous, thoughtful, interesting, and this topic is important. Although the novel methods are interesting and

appear to have merits, I am concerned that these new methods might overly complicate the overall message of the paper. It is also not clear why these methods are needed when there are already a range of similar methods available that might be able to achieve the same purpose in a manner that might be easier for a typical reader to digest. Nonetheless, I think this paper is interesting and important.

Response:

We thank the reviewer for describing our work as interesting, important and for noting the high quality of our mass cytometry data. To address the concern that the new computational methods might seem to over complicate the study for a typical reader to digest, we have incorporated changes in the text throughout the paper sections to improve clarity on why we chose specific approaches and methods. Please also see our specific responses (below) that highlight reasons for inclusion of our new analysis methods here.

Specific comments:

Comment:

How does TRACER compare to other trajectory mapping methods? Why was it important to develop this method for this project? Minor: given that this project is related to cancer, I wonder if the authors should be concerned about the similarity of this name with TRACERx, an ongoing research project on the study of cancer evolution.

Response:

TRACER is a stochastic approach for reconstructing trajectories from single cell data. In contrast, most trajectory methods using single cell data (e.g. Wanderlust, Monocle^{9,12}) are deterministic. They impose “pseudotime” ordering of cells based on prior knowledge. These trajectory methods order cells onto a connected path according to an underlying dynamic process that may or not include a bifurcation point. This is evident from the Slingshot analysis we ran on our data, shown in our revised manuscript in Supplementary Figure 6B. Given that EMT is believed to exhibit plasticity, we were motivated to develop a method that could give more than one possible trajectory cells may follow during transition. To this end, TRACER does not rely on pseudotime assumptions, and it assumes that EMT and MET have Markovian properties. Thus, TRACER presents EMT and MET state transitions as stochastic processes. The states in our case were derived through our CCAST analysis, and not during the TRACER analysis. In the end through bootstrap analysis, TRACER generates a distribution of the transition probabilities between EMT and MET states to assess differences between EMT and MET transitions in a statistically rigorous manner. Therefore, TRACER’s importance lies in the fact that it presents more than one possible EMT and MET trajectory (Figure 4F), based on the observations that result from the proportion of cells in each state at each time point. One could argue that TRACER is better suited for processes like EMT that is characterized by plasticity and bi directionality as opposed to other trajectory methods that are usually used for developmental processes that are mostly viewed as uni-directional. Nevertheless, our method does have limitations, since it does not take into account the expression of markers that regulate for example cell cycle and cell death, and these would offer insights into how each state functions/transitions with time. This is something we are working on for a new analysis. We have added text in our paper to improve clarity on why we chose the specific approach and methods used in this study, as well as discussion of the limitations mentioned above.

We thank the reviewer for bringing to our attention the use of the name TRACERx but hope that the confusion will be mitigated because TRACERx is a clinical study and TRACER is an algorithm.

Comment:

It is appreciated that several marker raw data shown. Is this low staining expected for slug and snail? Was staining for these confirmed with other cells. About pSMAD2/3 staining, these plots make it impossible to see the baseline data. Is it correct to say that pS6 was negative in the M* state given the high staining shown in Fig. 2. Perhaps change to relatively lower staining?

Response:

Although we were not able to detect basal (or TGFβ -induced) Slug and Snail protein levels in HCC827 cells, this could be a result of our specific culture conditions or simply a matter of the time-points we chose to analyze. For example, we cannot exclude the possibility that these may be activated in earlier EMT time points not tested here, as

suggested elsewhere¹³. Even though there is literature that supports our findings^{14,15}, there are also studies that claim that HCC827 cells express these transcription factors^{16,17}. Generally, we would not expect epithelial cells to express basal levels of EMT transcription factors. Furthermore, it is believed that there are various mechanisms of EMT induction, meaning that cells may express specific EMT transcription factors depending on the conditions/EMT inducers used (e.g. hypoxia, drugs, microenvironment¹⁸). In support of this, we present data below (from our lab but not related to this paper) where we detected significant Snail levels with CyTOF when we co-culture HCC827 cells with cancer-associated fibroblasts (below), as oppose to HCC827 cells in basal conditions or in presence of TGF β . Fibroblasts have been reported to induce EMT in cancer cells in paracrine fashion¹⁹. As for Slug, we have not yet been able to detect this factor in HCC827 cells, even though we have detected it in other cells. Below we show that when we used the same clone Slug antibody for western blotting, HCC827 cells showed extremely low levels compared to HCC1806 cells that have been reported to express Slug²⁰.

(A) Immunoblotting of HCC1806 and HCC827 lysates probed with the same clone Slug antibody used for mass cytometry (CST, C19G7). (B) Mass cytometry measurements of Snail protein in various HCC827 samples: Untreated or basal (Ctrl), Treated with TGF β for 6 days, and co-cultured with cancer associated fibroblasts for 24 hrs

To help visualize pSMAD basal levels, we have added to Supplementary Figure 2 of the revised manuscript, differently oriented histograms to show basal levels of not only pSMAD, but also Vimentin, CD44 and Twist, all low in basal conditions (Supplementary Figure 2D). As for pS6 in the M* state, we agree, that it is incorrect to use the term “negative”. We have changed this to “low” staining in the revised manuscript.

Comment:

What is the basis for choosing the CCAST clustering algorithm. This does appear to be an interesting method but it is insufficiently introduced in the introduction. Some brief description of what this does would be appreciated.

Response:

CCAST is published and available as a R package. CCAST³ combines clustering with a statistical framework based on non-parametric tests (i.e. no assumption) to partition the data, corrected for multiple testing to avoid overfitting. This approach results in unbiased marker selection and in general does not require pruning. In this study, CCAST not only identifies relevant clustering markers in an unbiased manner, but it also offers refinement of clusters through recursive partitioning, a well-established technique that correctly classifies members of certain populations based on several dichotomous dependent variables in the form of a decision tree. Furthermore, the hierarchical nature of data partitioning allows for unbiased ranking of clusters based on a predefined number of subpopulation parameters e.g. minimum number of cells. We have added additional information in our paper to introduce CCAST and the basis for choosing it for our analysis.

Comment:

It seems like the gating strategies produced by this algorithm could have been used in place of the STAMP method that the authors have developed. What is important about the STAMP method that other metaclustering approaches? Did the authors simply trying using the gating strategies defined by CCAST on each of the samples to assess the relative contributions of each of these clusters?

Response:

The EMT-MET State MaP (STAMP) (changed in the revised manuscript to EMT-MET PHENotypic State MaP (PHENOSTAMP)), is the result of a combination of methods we used to 1) define EMT and MET clusters (i.e. states) with CCAST, 2) construct a reference map of these states to then 3) project on this map cell lines and clinical specimens to phenotype them towards their EMT status. In detail, following clustering with CCAST, we generated a

2D t-SNE projection of our derived EMT and MET states from our mass cytometry study. Next, we defined regions of the map most uniquely associated with each state and determined the highest density region per state per sampling time and estimated its respective centroid. We then employed Voronoi and Convex Hull analysis to achieve density-driven segmentation of the EMT t-SNE landscape, thus constructing the reference EMT-MET PHENOSTAMP. The combination of the described computational steps above for building the map and for subsequent projections, are available in code and this has been provided for review under the name STAMP (will be changed to PHENOSTAMP). The reviewer is correct in that PHENOSTAMP is reliant on CCAST analysis. Basically, we are able to assess relative contributions of each of the CCAST-defined EMT-MET states in other cell lines and clinical specimens through projection onto the EMT-MET PHENOSTAMP using a neural net algorithm. We have incorporated changes in the text throughout the paper sections to improve clarity on this matter.

Comment:

Related to Supplementary Table 2, please show principal component loadings in addition to these p values.

Response:

The principal component loadings have been added in Supplementary Figure 3 in the revised manuscript.

Comment:

Although the force directed layout plots are very nice. This method introduces bias into the visualization by incorporating information about sample order. Did the authors compare to other analyses that are typically used for visualizing trajectories such as diffusion map or UMAP?

Response:

In this study we not only wanted to define EMT and MET states at the single-cell proteomic level, but we also wanted to visualize them by incorporating time information (and sample order) to provide deeper insight on EMT and MET spatiotemporal dynamics. Our Vortex²¹ force-directed layouts were produced similar to the way presented in Zunder et al. with the use of FLOW-MAP²². Because this is a layout often used, we included for readers who are familiar with this type of visualization of time-resolved data. We also felt that this visualization approach provides better insight on what happens when cells return to their basal phenotypic states after undergoing MET. However, we also included a t-SNE visualization, as this is commonly applied as well. The t-SNE visualization is similar to a UMAP visualization (see below). Unlike FLOW-MAP, both t-SNE (and UMAP) visualizations are appealing because they embed the data into a metric space. Metric space is essential to our study because our ultimate goal was to project new samples onto the states defined by our time course in-vitro analysis.

While UMAP appears less noisy (as it is designed to do), it seemed to us that it introduced an artifact that is not consistent with the temporal observations as well as the TRACER results concerning the hysteresis phenomenon we reported. In particular, UMAP places the MET state seemingly between the pEMT and M states, yet during EMT, the MET state is not visited and does not precede the M state as do the pEMT states.

Comparison of t-SNE and UMAP organization of the 8 EMT and MET states

Comment:

The last sentence of the abstract is an overstatement given the small number of samples analyzed and the lack of any statistical analysis this should be modified to be more accurate.

Response:

We have modified the last sentence of the abstract to better reflect our study and our drawn conclusions: "In summary, we provide a framework to phenotypically characterize clinical samples in the context of in vitro EMT-MET studies that, through future studies, may be used to assess metastasis and drug sensitivity"

Comment:

About software availability, given that there are a number of less standard and novel analysis tools used here and in various combinations, I think that software or code used to generate the data should be made available.

Response:

We have provided our code for review with the manuscript. In the revised manuscript we have added the statement under DATA AND SOFTWARE AVAILABILITY: Mass cytometry time-course data will be made available through Cytobank. All codes and related software will be available on Github.

Reviewer #4 (Remarks to the Author):

Comment:

In general I think this is a very worthwhile study. First, there has been a relative paucity of time-resolved studies of EMT and (especially) MET; the typical data in the field have only two or three time points. Second, the emphasis on single-cell measurements is critical if one wants to disentangle any process that is likely to involve heterogeneous populations of cells. In general I found the data quite believable (and I was happy with the attention to such details as the effects of confluency) and the analysis methods quite informative. Thus, in my opinion this work deserves publication.

Response:

We thank Dr. Levine for his supportive assessment of our work and his positive feedback and recognition of the novel aspects of our study. We agree that incorporating multiple time points in combination with multiplexing the plethora of disparate markers for interrogating the EMT-MET system at the single-cell level offers a perspective of this biological process that has not been previously captured. To address the comments below, we have incorporated changes to improve text clarity in the paper and we have added the references that have been suggested.

Comment:

Of course I did have some requests for clarification of various points that I found unclear. First, the repeated use of the word "continuum": when talking about the state space was confusing. The authors find a finite number of states (eight for their detailed mass-spec data with "minimal within-state marker heterogeneity" (p.8)). So, why do they conclude that there is no useful way to identify discrete states and that the states are best thought of as forming a continuum? This is a topical question as most recent studies argue in favor of some relatively small number of major phenotypes (see Patsushenko et al, Nature 556 (7702), 463 (2018); Jia, Dongya, et al. "Testing the gene expression classification of the EMT spectrum." Physical biology (2019)) whereas some modeling studies suggest something like a continuum (Font-Clos, Francesc, Stefano Zapperi, and Caterina AM La Porta. "Topography of epithelial-mesenchymal plasticity." Proceedings of the National Academy of Sciences 115.23 (2018): 5902-5907) - perhaps this issue should be discussed in greater detail and some of these references added.

Response:

We appreciate Dr. Levine's comment here and added the recommended references in our revised manuscript.

The issue continuous vs discrete set of states is indeed an important one, and we should be more clear and careful with our language here. The bottom line is that we cannot ascertain whether the EMT changes are continuous or discrete. Even though we have measured more time points and markers at the single-cell level than previously reported in EMT studies, our measurements are in fact at discrete time-points and we report a discrete set of states. We clarify this point in the text and chose to replace the word “continuum” with “spectrum”, which we feel may be more appropriate.

Interestingly, observation of our protein data support the notion that EMT and MET states may form a continuum. In particular, our biaxial protein expression plots of the canonical markers (Figures 1 & 2) suggest a continuum of states. Also, our t-SNE plots in Figure 4 and our clinical specimen projections in Figure 6 show cells occupying continuous, and not distinct, disconnected regions of the map. However, one needs to be careful in over interpreting the visualization outputs, given that t-SNE performs nonlinear transformation of the data.

Much like other cell differentiating systems, what is actually a continuous process is still often broken up into discrete landmarks for descriptive and summary purposes. To this point, our analysis is focused on computationally defining discrete states based on the high dimensional nature of the data in effort to better characterize the gradual processes of EMT and MET. With this in mind we have incorporated the suggested changes throughout the text to better discuss and clarify this important issue.

Comment:

Another point where the discussion is perhaps over-simplified is the initial statements and subsequent validation of the STAMP approach using for example A549 cells. As far as I can tell the A549 data was not single cell and therefore one cannot conclude that all the cells are in a pEMT state. In fact previous studies of the EMT spectrum of A549 cells (George, Jason T., et al. "Survival outcomes in cancer patients predicted by a partial EMT gene expression scoring metric." *Cancer research* 77.22 (2017): 6415-6428 - see Figure 3 therein) argue that A549 is rather inhomogeneous, with some pEMT cells but many E and M as well; averaging this over the population by making bulk measurements is rather misleading. I was very happy to see that the comparison with the patient samples was done at the single cell level and does not suffer from this confusion.

Response:

We apologize for any misunderstanding and have clarified this in the revised manuscript. While our initial estimations of A549 EMT phenotypic properties in Figure 1 were made through bulk analysis, most of our analysis on A549 cells was at the single-cell level. Specifically, we generated (single-cell) mass cytometry data of A549 cells and then projected on the map to assess their EMT-MET phenotypic states (Figure 5). Using our refined definitions and single-cell analysis, we found that the majority of A549 cells in basal conditions appear to be in a pEMT state (in our case, pEMT3 specifically), and a significant proportion of them in the M state. We clarified this point in the respective results section as well as in the Figure 5 figure legend. Furthermore, we provide additional mass cytometry data in Supplementary Figure 7 of the revised manuscript where we show the raw expression data of all cellular markers analyzed in the A549 and H3255 cell lines (that were subsequently projected on our EMT-MET state map shown in Figure 5).

Comment:

The authors take as given the somewhat standard view that stem cells exhibit a Cd24-CD44+ signature. This appears to be correct for mesenchymal stem cells, but recent work in the breast cancer context (Grosse-Wilde, Anne, et al. "Stemness of the hybrid epithelial/mesenchymal state in breast cancer and its association with poor survival." *PLoS one* 10.5 (2015): e0126522; Goldman, Aaron, et al. "Temporally sequenced anticancer drugs overcome adaptive resistance by targeting a vulnerable chemotherapy-induced phenotypic transition." *Nature communications* 6 (2015): 6139 has suggested that there can one stem-like behavior associated with pEMT and with CD24+CD44+ markings.

Response:

We appreciate this being brought to our attention. Although we are aware of studies that show stemness of pEMT cells, we didn't functionally test “stemness” of the various states in our study, and therefore only discussed what the “standard view” is for the CD24-CD44 + signature. However, it was an omission to not elaborate further on pEMT and stemness, and thus we have incorporated in our discussion a section on this along with the suggested references.

Comment:

The results about hysteric features of the E->M->E loop and the new type of pEMT state are quite important and as argued tend to agree with previous arguments, albeit with a lot better data here. It would be worth highlighting a bit more the idea that is partially mentioned here that the return trajectory might depend on exactly how long the TGF-beta stimulation was applied for. Here it seems that applying it for less time makes return (mostly from pEMT states) more reversible, but the same might be true for much longer stimulation protocols. This may be related to the percentage of cells that do not return to E under this protocol. This idea may eventually merge into results for MET on sarcoma cell lines (which have been M for a very long time) - see Somarelli, Jason A., et al. "Mesenchymal-epithelial transition in sarcomas is controlled by the combinatorial expression of MicroRNA 200s and GRHL2." *Molecular and cellular biology* 36.19 (2016): 2503-2513.

Response:

We thank Dr. Levine for bringing to our attention additional literature on the matter of hysteresis. We have added text to highlight and discuss further the idea of duration of TGFb stimulation and hysteresis, while also incorporating the reference mentioned above.

Comment:

Unless I am missing something, there appears to be some confusion in Figure numbers on p 6 (lines 120-125). Could the authors check these?

Herbert Levine

Response:

The error pertained to Figure numbers on page 6 have been corrected in the revised manuscript.

References

1. Mani, S. A. *et al.* The Epithelial-Mesenchymal Transition Generates Cells with Properties of Stem Cells. *Cell* **133**, 704–715 (2008).
2. Pastushenko, I. *et al.* Identification of the tumour transition states occurring during EMT. *Nature* **556**, 463–468 (2018).
3. Anchang, B., Do, M. T., Zhao, X. & Plevritis, S. K. CCAST: A Model-Based Gating Strategy to Isolate Homogeneous Subpopulations in a Heterogeneous Population of Single Cells. *PLOS Comput. Biol.* **10**, e1003664 (2014).
4. Kalluri, R. & Weinberg, R. A. The basics of epithelial-mesenchymal transition. *J. Clin. Invest.* **119**, 1420–1428 (2009).
5. Ramlau, R. *et al.* A Phase II Study of Tg4010 (Mva-Muc1-II2) in Association with Chemotherapy in Patients with Stage III/IV Non-small Cell Lung Cancer. *J. Thorac. Oncol.* **3**, 735–744 (2008).
6. Fanni, D. *et al.* MUC1 in mesenchymal-to-epithelial transition during human nephrogenesis: changing the fate of renal progenitor/stem cells? *J. Matern. Fetal Neonatal Med.* **24**, 63–66 (2011).

7. Wang, J. *et al.* Loss of Trop2 Promotes Carcinogenesis and Features of Epithelial to Mesenchymal Transition in Squamous Cell Carcinoma. *Mol. Cancer Res.* **9**, 1686–1695 (2011).
8. Bendall, S. C. *et al.* Single-Cell Mass Cytometry of Differential Immune and Drug Responses Across a Human Hematopoietic Continuum. *Science* **332**, 687–696 (2011).
9. Bendall, S. C. *et al.* Single-Cell Trajectory Detection Uncovers Progression and Regulatory Coordination in Human B Cell Development. *Cell* **157**, 714–725 (2014).
10. Behbehani, G. K., Bendall, S. C., Clutter, M. R., Fantl, W. J. & Nolan, G. P. Single-cell mass cytometry adapted to measurements of the cell cycle. *Cytometry A* **81A**, 552–566 (2012).
11. Tapia, C. *et al.* Two mitosis-specific antibodies, MPM-2 and phospho-histone H3 (Ser28), allow rapid and precise determination of mitotic activity. *Am. J. Surg. Pathol.* **30**, 83–89 (2006).
12. Trapnell, C. *et al.* The dynamics and regulators of cell fate decisions are revealed by pseudotemporal ordering of single cells. *Nat. Biotechnol.* **32**, 381–386 (2014).
13. Choi, J., Park, S. Y. & Joo, C.-K. Transforming Growth Factor- β 1 Represses E-Cadherin Production via Slug Expression in Lens Epithelial Cells. *Invest. Ophthalmol. Vis. Sci.* **48**, 2708–2718 (2007).
14. Chang, T.-H. *et al.* Slug Confers Resistance to the Epidermal Growth Factor Receptor Tyrosine Kinase Inhibitor. *Am. J. Respir. Crit. Care Med.* **183**, 1071–1079 (2011).
15. Lee, A.-F. *et al.* Reverse epithelial-mesenchymal transition contributes to the regain of drug sensitivity in tyrosine kinase inhibitor-resistant non-small cell lung cancer cells. *PLOS ONE* **12**, e0180383 (2017).
16. Lu, Y., Liu, Y., Oeck, S. & Glazer, P. M. Hypoxia Promotes Resistance to EGFR Inhibition in NSCLC Cells via the Histone Demethylases, LSD1 and PLU-1. *Mol. Cancer Res.* **16**, 1458–1469 (2018).
17. La Monica, S. *et al.* Combination of Gefitinib and Pemetrexed Prevents the Acquisition of TKI Resistance in NSCLC Cell Lines Carrying EGFR-Activating Mutation. *J. Thorac. Oncol.* **11**, 1051–1063 (2016).
18. Cursons, J. *et al.* Stimulus-dependent differences in signalling regulate epithelial-mesenchymal plasticity and change the effects of drugs in breast cancer cell lines. *Cell Commun. Signal.* **13**, 26 (2015).
19. Laberge, R.-M., Awad, P., Campisi, J. & Desprez, P.-Y. Epithelial-Mesenchymal Transition Induced by Senescent Fibroblasts. *Cancer Microenviron.* **5**, 39–44 (2012).

20. Dang, T. T., Esparza, M. A., Maine, E. A., Westcott, J. M. & Pearson, G. W. $\Delta Np63\alpha$ Promotes Breast Cancer Cell Motility through the Selective Activation of Components of the Epithelial-to-Mesenchymal Transition Program. *Cancer Res.* **75**, 3925–3935 (2015).
21. Samusik, N., Good, Z., Spitzer, M. H., Davis, K. L. & Nolan, G. P. Automated mapping of phenotype space with single-cell data. *Nat. Methods* **13**, 493–496 (2016).
22. Zunder, E. R., Lujan, E., Goltsev, Y., Wernig, M. & Nolan, G. P. A Continuous Molecular Roadmap to iPSC Reprogramming through Progression Analysis of Single-Cell Mass Cytometry. *Cell Stem Cell* **16**, 323–337 (2015).

REVIEWERS' COMMENTS:

Reviewer #1 (Remarks to the Author):

The stated goal of this study is to provide a framework to phenotypically characterize clinical samples with respect to EMT-MET status that may be used to assess metastasis and drug sensitivity. My prior concern and skepticism regarding the clinical application of the findings remains rather unchanged. The authors' hope that by phenotyping clinical samples in terms of EMT and MET dynamic state changes with single-cell resolution, the findings would impact treatment strategies seems rather remote at best given issues with tumor sampling through biopsies, the trend toward targeted therapeutics based on driver oncogenes and the current interest in immunotherapy all of which are not informed by EMT status.

Having made these statements regarding practical clinical implications, the study still provides some biological insights regarding EMT states at the single cell level and the authors have responded to reviewer comments with clarifications and editorial changes. Perhaps a validation study of the findings based on analysis of an additional tumor set or multiple biopsy specimens from given subjects or validation in an additional independent tumor type would alleviate some of the concerns regarding the significance of the findings.

Reviewer #3 (Remarks to the Author):

My concerns have been adequately addressed. Thanks for these clarifications.

Reviewer #4 (Remarks to the Author):

The authors have addressed all my requests for clarification and additional discussion of some of the important points raised by their data. I am now in favor of publication.

The authors would like to thank the reviewers and the editor for their time and for providing us with their thoughtful and beneficial critiques of our manuscript during the second round of revisions.

After studying the reviewers' comments, the following point-by-point responses have been prepared:

Reviewer #1 (Remarks to the Author):

Comment:

The stated goal of this study is to provide a framework to phenotypically characterize clinical samples with respect to EMT-MET status that may be used to assess metastasis and drug sensitivity. My prior concern and skepticism regarding the clinical application of the findings remains rather unchanged. The authors' hope that by phenotyping clinical samples in terms of EMT and MET dynamic state changes with single-cell resolution, the findings would impact treatment strategies seems rather remote at best given issues with tumor sampling through biopsies, the trend toward targeted therapeutics based on driver oncogenes and the current interest in immunotherapy all of which are not informed by EMT status.

Having made these statements regarding practical clinical implications, the study still provides some biological insights regarding EMT states at the single cell level and the authors have responded to reviewer comments with clarifications and editorial changes. Perhaps a validation study of the findings based on analysis of an additional tumor set or multiple biopsy specimens from given subjects or validation in an additional independent tumor type would alleviate some of the concerns regarding the significance of the findings.

Response:

We agree with the Reviewer that we did not prove that our EMT state mapping is a clinically utility tool in our current study. Instead, we introduced the key concepts for developing the EMT state map using a single-cell in vitro time-series dataset and showed initial applications to clinical samples. Proving clinical utility is beyond the current scope of our work because doing so will require a prospective collection of fresh human tumor samples for single-cell dissociation (as we have done with the clinical samples presented). In addition, such a prospective study will require significant follow-up to assess clinical outcomes. That said, we are moving toward that ultimate goal. To address any of the reviewer's concerns that we may be overemphasizing the clinical relevance of our work in the current manuscript, we toned down the discussion of clinical relevance. However, we think it is important to emphasize that existing assays are not sufficient for treatment planning and predicting drug resistance. For example, targeting therapeutics based on driver oncogenes alone and/or immunotherapy is a great advancement but leaves unanswered many mechanisms of drug resistance. Increasing evidence demonstrates that cellular plasticity contributes to drug resistance. In particular, several studies focus on the link between EMT and drug resistance and/or poor clinical outcomes¹⁻⁴. Finding new ways to assay plasticity in general, and EMT in particular, will be in the future paramount towards characterizing clinical relevance of EMT and in particular its role on drug resistance and response.

Reviewer #3 (Remarks to the Author):

Comment:

My concerns have been adequately addressed. Thanks for these clarifications.

Response:

We thank the reviewer for their support.

Reviewer #4 (Remarks to the Author):

Comment:

The authors have addressed all my requests for clarification and additional discussion of some of the important points raised by their data. I am now in favor of publication.

Response:

We thank the reviewer for their support.

References

1. Zheng, X. *et al.* Epithelial-to-mesenchymal transition is dispensable for metastasis but induces chemoresistance in pancreatic cancer. *Nature* **527**, 525–530 (2015).
2. Byers, L. A. *et al.* An Epithelial–Mesenchymal Transition Gene Signature Predicts Resistance to EGFR and PI3K Inhibitors and Identifies Axl as a Therapeutic Target for Overcoming EGFR Inhibitor Resistance. *Clin. Cancer Res.* **19**, 279–290 (2013).
3. Pattabiraman, D. R. & Weinberg, R. A. Targeting the Epithelial-to-Mesenchymal Transition: The Case for Differentiation-Based Therapy. *Cold Spring Harb. Symp. Quant. Biol.* **81**, 11–19 (2016).
4. Tan, T. Z. *et al.* Epithelial–mesenchymal transition spectrum quantification and its efficacy in deciphering survival and drug responses of cancer patients. *EMBO Mol. Med.* **6**, 1279–1293 (2014).